# ACCELERATED CO-DESIGN OF ROBOTS THROUGH MORPHOLOGICAL PRETRAINING

**Luke Strgar and Sam Kriegman**
Northwestern University

## ABSTRACT

The co-design of robot morphology and neural control typically requires using reinforcement learning to approximate a unique control policy gradient for each body plan, demanding massive amounts of training data to measure the performance of each design. Here we show that a universal, morphology-agnostic controller can be rapidly and directly obtained by gradient-based optimization through differentiable simulation. This process of morphological pretraining allows the designer to explore non-differentiable changes to a robot's physical layout (e.g. adding, removing and recombining discrete body parts) and immediately determine which revisions are beneficial and which are deleterious using the pretrained model. We term this process "zero-shot evolution" and compare it with the simultaneous co-optimization of a universal controller alongside an evolving design population. We find the latter results in *diversity collapse*, a previously unknown pathology whereby the population—and thus the controller's training data—converges to similar designs that are easier to steer with a shared universal controller. We show that zero-shot evolution with a pretrained controller quickly yields a diversity of highly performant designs, and by fine-tuning the pretrained controller on the current population throughout evolution, diversity is not only preserved but significantly increased as superior performance is achieved. Videos and code can be found at this website.

## 1 INTRODUCTION

The co-design of morphology and control in robots is important because robots perform better when their physical layout is optimized for their intended niche—like a fish out of water, a good body in one domain can obstruct the acquisition of intelligent behavior in another, if it is unable to evolve. However, over the past three decades of research, despite exponential increases in computing power, surprisingly little tangible progress has been achieved beyond the very first co-designed robots (Sims, 1994). This stagnation is due in part to the nested complexity of evolving a robot's morphology and learning a bespoke controller for every morphological variant. Because controllers are usually optimized in non-differentiable simulations using reinforcement learning (RL), large amounts of training data are needed to effectively learn a single controller for a single morphology, a cost that is compounded by repeatedly relearning new controllers as the robot's morphology changes throughout evolution.

As a result, the overwhelming majority of prior work has been limited to small numbers of morphologically simple robots that exhibit simple behaviors in simple environments. Even with simplifying assumptions that significantly speed simulation, such as constraining the design space to infinitely rigid "stick figures" composed of less than a dozen body parts, there is usually only enough time to explore a few thousand morphologies (Wang et al., 2019; Zhao et al., 2020; Gupta et al., 2021; Yuan et al., 2022; Lu et al., 2025; Yu et al., 2026). Others have relaxed this constraint by considering more flexible bodies composed of many deformable cells (Cheney et al., 2018; Kriegman et al., 2020a; Li et al., 2025), but due in part to the increased computational burden of simulating soft materials, these robots have had lower motoric complexity (fewer independent motors) and have been much less intelligent (completely unresponsive to external stimuli) compared to their rigid-bodied counterparts. Often the robots are restricted to two dimensional worlds (Medvet et al., 2021; Wang et al., 2023b; Strgar et al., 2024). (See Appx. A.1 for more detailed discussion of this related work.)

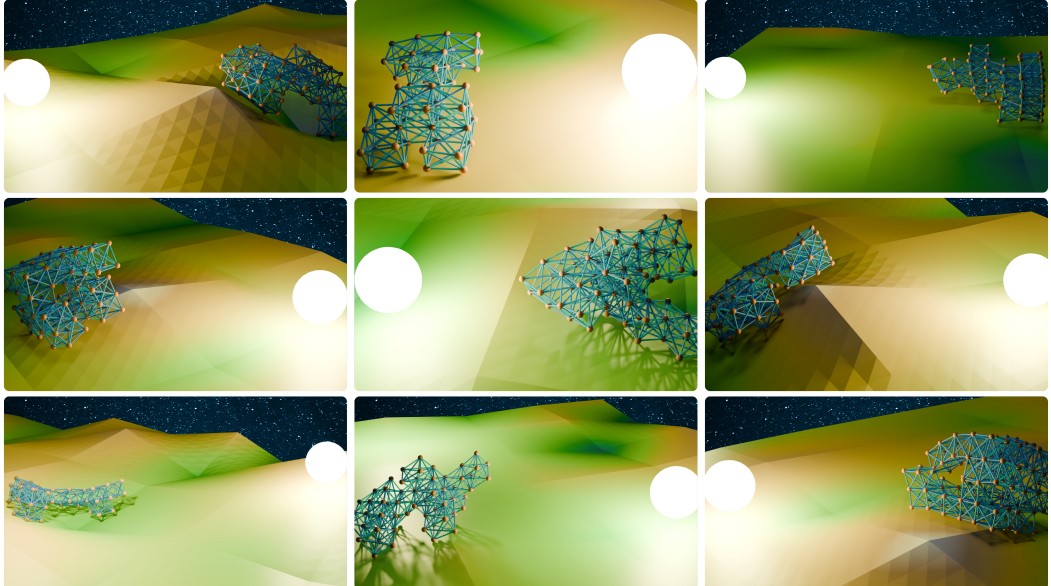

Figure 1: **Universal control of differentiable robots.** Large-scale pretraining and fine-tuning of a universal controller was achieved by averaging simulation gradients across the robot's body, world, and goal. The controller is shared by an arbitrarily large and morphologically diverse population of robots as they undergo morphological evolution. The objective is to find designs that can move quickly across a previously-unseen terrain toward a randomly positioned light source.

Inspired by the remarkable success of large-scale pretrained models in computer vision and natural language processing, we pretrain a universal controller across millions of complex body plans using gradient information from differentiable simulation, averaging gradients across variations in the robot's body, world and goal (Fig. 1). Armed with a universal controller, evolution can now iteratively improve the robot's morphology, and the controller can be rapidly fine-tuned for the current population with simulation gradients (Fig. 2). This also enables the successful recombination of designs (a.k.a. crossover; Fig. 4), a hallmark of biological evolution and of human engineering that has yet to be convincingly demonstrated in robots.

Indeed there is a tacit assumption in robotics that crossover—the combining of two parent designs to produce offspring—is so unlikely to produce viable offspring, that it is better to omit crossover altogether and focus entirely on small mutations that slightly alter a single design parent to produce offspring. While instances have been reported in which two morphologies were combined using crossover to produce a new morphology (Sims, 1994; Bongard & Pfeifer, 2001; Hiller & Lipson, 2010; Strgar et al., 2024), it was not clear if crossover ever produced offspring with equal or better fitness than either one of their parents—or if the recombined designs were even better than randomly generated robots. Here we show how a pretrained universal controller can unlock successful crossover of robots.

Several cases have been reported in the literature in which RL was used to approximate a universal control policy gradient across a small dataset of previously-designed (Huang et al., 2020; Gupta et al., 2022; Xiong et al., 2023; Bohlinger et al., 2024) or simultaneously co-designed (Schaff & Walter, 2022; Wang et al., 2023b; Li et al., 2025) morphologies. However, the inefficiencies of policy training without recourse to gradient information precluded large-scale pretraining. As we detail below, co-designing morphology and universal control simultaneously from scratch can, and without careful consideration almost certainly will re-

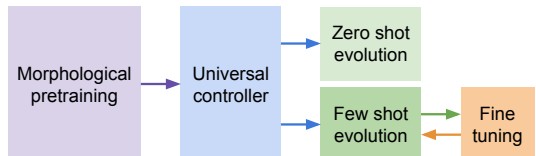

Figure 2: **Overview of the proposed method.** End-to-end differentiable policy training across tens of millions of morphologically distinct robots—morphological pretraining—produces a universal controller, which is kept frozen throughout zero-shot evolution and fine-tuned for each generation of few-shot evolution.

sult in diversity collapse, inhibiting co-design by reducing it to policy training for a single design. Others (Ma et al., 2021; Matthews et al., 2023; Cochevelou et al., 2023; Strgar et al., 2024; Kobayashi et al., 2024) have utilized first-order gradients from differentiable simulation to speed co-design. But, a custom controller still needed to be learned for each morphology, and the resulting robots could only exhibit rote behaviors, such as locomotion in a straight line. (See Appx. A.1.)

Here we demonstrate a more scalable approach that starts with large-scale morphological pretraining in differentiable simulation and yields a single morphology-agnostic controller capable of adaptive sensor-guided behavior across evolving populations of distinct, complex mass-spring robots with hundreds to thousands of independent motors.

## 2 METHODS

In this section we describe the search space ("morphospace"; Sect. 2.1), the simulator (Sect. 2.2), the universal controller (Sect. 2.3), morphological pretraining (Sect. 2.4), zero-shot evolution (Sect. 2.5), few-shot evolution with generational fine-tuning (Sect. 2.6), and simultaneous co-design (Sect. 2.7).

### 2.1 MORPHOSPACE

Robot morphologies were genetically encoded as contiguous collections of voxels within a $6 \times 6 \times 4$ (Length $\times$ Width $\times$ Height) binary genotype workspace, $\mathcal{G}$. Voxelized genotypes were then mapped to a phenotype space $\mathcal{P}$ comprising masses $\mathcal{M}$ and springs $\mathcal{S}$ arranged in a cubic lattice with 10 cm$^3$ unit cells (Fig. 3). More specifically, a genotype voxel at position $(i, j, k)$ in $\mathcal{G}$ is expressed phenotypically by eight masses, one in each corner of the corresponding cubic cell in $\mathcal{P}$ with coordinates $(0.1i+\delta_x, 0.1j+\delta_y, 0.1k+\delta_z)$ where $\delta_{x,y,z} \in \{0, 0.1\}$. Springs are then connected to these masses in two patterns: (1) axial springs along cube edges, and (2) planar diagonal springs in each face. Adjacent genotype voxels share masses and springs at their interfaces, ensuring that contiguous structures in $\mathcal{G}$ map to cohesive mass-spring networks in $\mathcal{P}$.

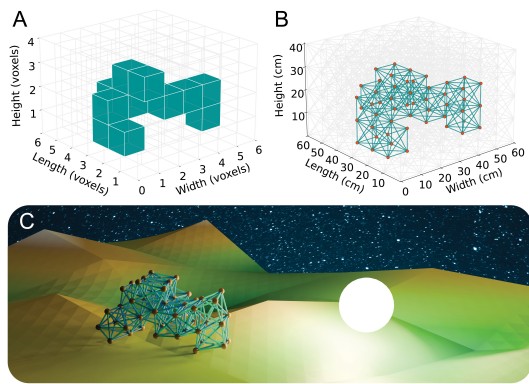

Figure 3: **Genotype to phenotype.** Designs are encoded by a voxel genotype (**A**), which is expressed as a spring-mass phenotype (**B**), and evaluated in a differentiable environment (**C**). Springs (teal lines in B and C) and masses (orange spheres) are motorized and sensorized, respectively.

The resultant $6 \times 6 \times 4$ workspace accommodated a maximum of $|\mathcal{M}| = 245$ potential mass positions and $|\mathcal{S}| = 1648$ potential springs. Each robot was centered in the x-y plane according to its center of mass and shifted to the bottom of the workspace to ensure ground contact prior to behavior. This procedure ensured stable initial conditions for locomotion while maintaining consistent relative positioning between robots of different morphologies.

To identify unique morphologies, we defined an equivalence relation on the genotype space that accounted for translations and symmetries. Two genotypes were considered identical if, after aligning their occupied voxels to the origin, one could be transformed into the other through any combination of: (1) 90° rotations about the z-axis, (2) reflection about the x-axis, or (3) reflection about the y-axis. Each unique design was represented by its lexicographically minimal form across all such transformations.

### 2.2 DIFFERENTIABLE SIMULATION

We here extend the differentiable 2D mass-spring simulator developed by Strgar et al. (2024) to three dimensions and add exteroception: perception of external stimuli outside the body, namely light. Masses on $\mathcal{M}$ hosting photoreceptors were connected by actuating springs on $\mathcal{S}$ (defined in Sect. 2.1), which exerted forces on their endpoint masses to perform phototaxis.

During simulation, spring rest lengths may be actuated continuously between $\pm 20\%$ of their initial values derived from $\mathcal{P}$ (see Sect. 2.1). Spring forces were computed according to Hooke's law $F = k(L - L_0)$, where $k = 1.5 \times 10^4$ N/m is the spring stiffness coefficient, $L$ is the current spring length, and $L_0$ is the modulated rest length. Resulting impulses, as well as damping and gravitational forces, were used to update velocities for each mass, and in turn mass positions were updated using the new velocities.

The terrains along which robots behaved were modeled using randomly sampled height maps (see Appx. A.3 for details). During simulation, terrain heights at arbitrary coordinates $(x, y)$ were computed through bilinear interpolation of the height map. For collision handling, we detected when a mass's updated $z$-coordinate fell below the interpolated terrain height at its $(x, y)$ position. Upon detection, we employed a three-phase resolution: (1) iterative bisection on the interval $[0, dt]$ to estimate the time of impact and advance the mass to the contact point, (2) velocity projection onto the contact surface normal (estimated via central differences), and (3) constrained motion along the surface tangent for the remaining timestep. Following Strgar et al. (2024), friction forces were computed by negating the tangential velocity component and clamping its magnitude to not exceed the magnitude of the normal velocity component.

Our simulator was implemented in the Taichi programming language (Hu et al., 2020), providing both GPU acceleration for parallel, multi-robot simulation and automatic differentiation capabilities. The simulator was directly integrated with a PyTorch-based universal controller (Sect. 2.3), enabling end-to-end backpropagation through 1000 timesteps ($dt = 0.004$s) of physics simulation and neural control for gradient-based optimization of the controller parameters.

## 2.3 THE UNIVERSAL CONTROLLER

We employed a simple multi-layer perceptron (MLP) as a universal controller for adaptive phototaxis: guiding a population of thousands of morphologically diverse robots towards arbitrarily positioned light sources across randomly varying, rugged terrains. The network mapped two input streams to spring actuation signals: photosensor readings from masses and central pattern generator (CPG) inputs. To accommodate all possible body plans in the morphospace (defined in Sect. 2.1), the network's input dimension was set to $|\mathcal{M}|$ (the maximum number of masses) and output dimension to $|\mathcal{S}|$ (the maximum number of springs). Sensors and actuators not present in a specific robot's body had their corresponding signals masked to zero, providing an implicit morphological conditioning through observation and action space masking.

Each mass-bound photosensor measured light intensity, which decays as distance to the light source position increases (see Sect. A.5). Sensor readings for each robot were normalized by subtracting the mean computed across that robot's active (unmasked) sensors, providing a zero-centered, embodied irradiance gradient. Following Hu et al. (2020), CPG inputs contained five sinusoidal waves with angular frequency $\omega = 10$ rad/s and phase offsets evenly spaced by $2\pi/5$ radians. Over the 4 sec simulation period (1000 steps, $dt = 0.004$s), these oscillators completed approximately six cycles.

The MLP architecture consisted of an input layer (dim 250: $|\mathcal{M}|$ mass sensors plus 5 CPG inputs), three hidden layers (dim 256 each), and an output layer (dim 1648: $|\mathcal{S}|$ springs). Each hidden layer was followed by layer normalization and ReLU activation, while the output layer used a $\tanh$ activation. All layers included learnable biases. In total the model consisted of 620,912 learnable parameters. Network weights were initialized using a Xavier uniform distribution (gain=1.0) (Glorot & Bengio, 2010) with zero-initialized biases, and the network was optimized using Adam (Kingma & Ba, 2015) ($\beta_1 = 0.9$, $\beta_2 = 0.999$) with gradient norm clipping at 1.0. Learning rates were scheduled using variants of cosine annealing with restarts (detailed in Sects. 2.4, 2.6, and 2.7).

## 2.4 MORPHOLOGICAL PRETRAINING

The universal controller was pretrained across a dataset of over 10 million distinct robot morphologies (see Appx. A.2 for details). The controller was trained over 1400 learning steps to minimize the batch mean of $d_1/d_0$, where $d_1$ and $d_0$ represent each robot's final and initial distances from its target light source, respectively. This relative distance formulation ensured robots were not penalized for being initialized far from their targets and equally incentivized fine-grained control in robots initialized near their targets.

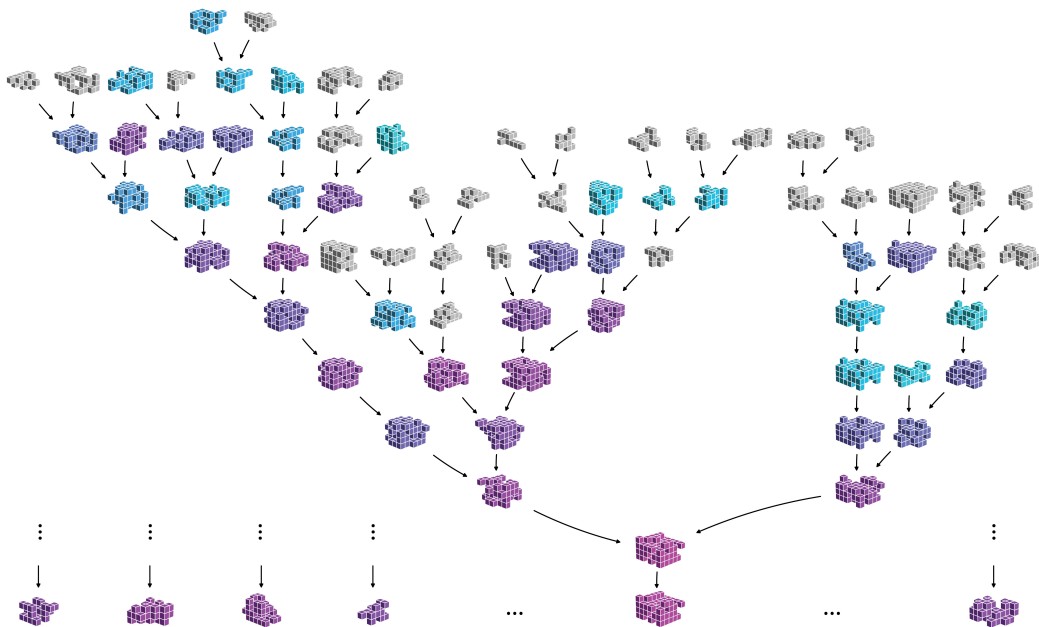

Figure 4: **Few-shot evolution.** A population of 8192 initially random designs (a pair of which are shown in the top row) were randomly recombined and mutated to produce 8192 offspring, temporarily expanding the population to 16384 designs. All designs in the population were driven by the same universal controller, which was rapidly pretrained (before evolution) and fine-tuned for the current population (at every generation of evolution) using analytical gradients from differentiable simulation. Deleting the worst performing designs and replacing them with the best offspring, and repeating this process for several generations, yields a diversity of increasingly performant designs, and ultimately a final population of 8192 winning designs (bottom row), each with their own unique evolutionary history (phylogeny). An example phylogenetic tree, colored by loss (decreasing from gray to cyan to pink), is shown for one of the winning designs.

We used a batch size of 8192, distributed in equal partitions of 1024 across a single compute node consisting of eight H100 SXM GPUs. Each sample consisted of a randomly-generated robot morphology (Appx. A.2), a randomly-generated terrain shape (Appx. A.3), and a randomly positioned light source (Appx. A.3), and was seen exactly once during training. Training used a cosine annealing with restarts schedule, with initial learning rate $1e^{-3}$, cycle length starting at 10 steps and doubling each restart, minimum learning rate $1e^{-5}$, and a decay rate of 0.7 applied to the starting learning rate at each cycle.

## 2.5 ZERO-SHOT EVOLUTION

Here, we introduce a novel robot design paradigm that leverages a frozen, pretrained universal controller to rapidly evaluate non-differentiable changes to a given robot's body plan. By using a single, fixed controller for all body plans, the design space may be efficiently explored without the computational burden of training a custom controller for each body plan. We refer to this method as "zero-shot evolution".

We initialized a population of 8192 random robot morphologies (unseen during pretraining) and evaluated each on a fixed test set of terrain and light source position pairs (see Appx. A.4 for details). A simple genetic algorithm was then applied iteratively: the population produced an equal number of offspring through two variation operators (described below), new offspring were evaluated once on the test set, and the top 50% across parents and offspring (using cached evaluation scores for parents) were selected to form the next generation.

Robot offspring were produced through one of two variation operators: mutation and recombination. The population was partitioned into two distinct groups: a random 25% of members were assigned to produce offspring through mutation, while the remaining 75% were reserved for producing off-

spring through recombination (or crossover). Each member in the mutation group produced a single offspring through random bit flip mutations performed on their genotype. Flips occurred with probability $p = 1/N$ where $N = 6 \times 6 \times 4$, the total number of voxels in the robot's genotype. After mutation, genotypes were processed to ensure validity: only the largest connected component was retained, and the resulting structure was translated to the bottom center of the workspace. If a mutation produced a body that was either empty or identical to a previously seen body, the process was repeated with the mutation rate increased by 2.5% until a valid, unique design was obtained.

From the recombination group (75% of the population), pairs of distinct parents were randomly sampled to produce offspring through crossover (Fig. 5). For each sampled pair, an offspring's genotype was created using a bitwise exclusive or (XOR) operation on the parent genotypes. As with mutation, post-processing retained only the largest connected component and centered it at the bottom of the workspace. If the resulting design duplicated a previous one, it was discarded. The sampling and generation process was repeated until the number of offspring equaled the size of the recombination group (75% of pop).

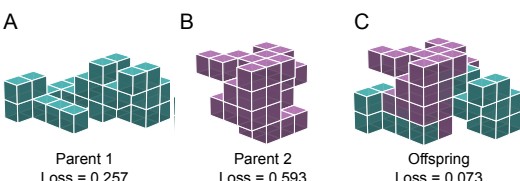

Figure 5: **Recombination of substructures.** A pair of designs (parents; **A, B**) is combined via crossover to produce a new design (offspring; **C**) that inherits components from both parents. The pretrained controller enabled several generations of successful recombination events such as this one.

## 2.6 FEW-SHOT EVOLUTION

In this experiment we extend the zero-shot paradigm (described above in Sect. 2.5) by fine-tuning the pretrained universal controller to the current population at every generation of morphological evolution. We refer to this approach as "few-shot evolution". The experimental setup of few-shot evolution matched the zero-shot case, with one key difference: before evaluation, each generation received 60 fine-tuning steps (30 for parents, 30 for offspring). The number of fine-tuning steps was empirically chosen to balance controller adaptation against evolutionary search while maintaining comparable wall-clock time across experiments. At the start of each generation, the controller's weights were reset to their pretrained values and the optimizer state was reinitialized. Fine-tuning used a cosine annealing learning rate schedule with initial and minimum rates of $3.5e^{-4}$ and $3.5e^{-5}$, respectively. The cycle length was set to 100; however, each cycle was truncated to align with one generation's 60 fine-tuning steps, resulting in an effective minimum learning rate of $1.5e^{-4}$. Since every generation re-initialized the pretrained weights, we did not decay the learning rate at the start of each cycle.

## 2.7 SIMULTANEOUS CO-DESIGN FROM SCRATCH

In our third and final experimental group, we remove morphological pretraining and instead simultaneously evolve a population of robots and learn their universal controller, from scratch. This is the algorithm used by Li et al. (2025). Here, it serves as a benchmark of the state of the art (bestowed with simulation gradients for fair comparison) while also isolating the effect of pretraining through ablation. Unlike few-shot evolution, controller parameters and optimizer state are inherited across generations rather than being reset. The genetic algorithm operates as before, but we reduce the per-generation training to just two learning steps (one for parents, one for offspring) to maintain parity with our pretraining experiments, where each training batch was unique.

Initially, we employed the same cosine annealing learning rate schedule used in morphological pretraining, but we found it was beneficial to reduce the start-of-cycle learning rate decay factor from 0.7 to 0.65 in order to stabilize learning across cycle restarts in this setting.

## 3 RESULTS

In this section we evaluate the results of morphological pretraining (Sect. 3.1), zero- and few-shot evolution (using the pretrained model; Sect. 3.2), and simultaneous co-design from scratch (without

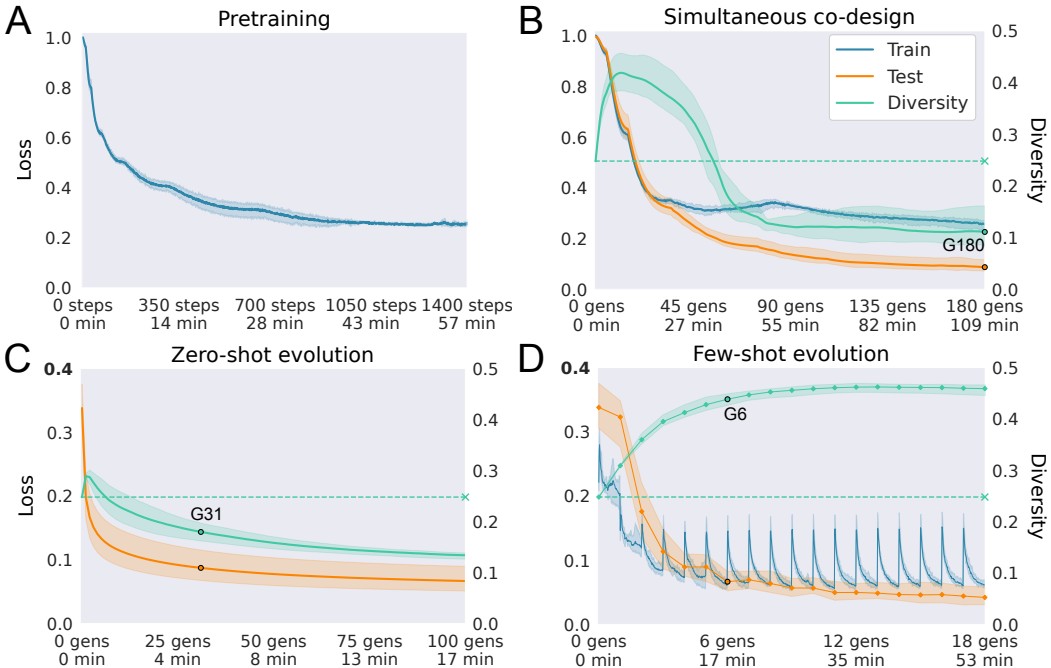

Figure 6: **Performance and diversity.** Morphological pretraining (**A**) converges with 70% improvement from baseline. The algorithm from Li et al. (2025), simultaneous co-design (from scratch without pretraining; **B**) achieves similar training loss; but, population diversity (mean pairwise Hamming distance on genotypes) collapses as evolution converges to a single species of similar designs which simplifies shared control. Zero-shot evolution (using the pretrained controller; **C**) rapidly improves test performance, but also suffers diversity collapse as evolution compiles slightly modified clones of the designs that are the most compatible with the pretrained model. Few-shot evolution (**D**) resets the pretrained controller at the start of each generation and performs 60 fine-tuning steps per generation. This significantly increases and sustains diversity as well as performance. Solid lines represent the batch (training; blue) or population (test; orange) means, averaged across three independent trials. Shaded regions surrounding the mean curves show the minimum and maximum values across the three trials. Note that pretraining loss (blue curve in A) is not directly comparable to co-design performance (B-D) because designs in the pretraining dataset only served as single-use training samples and were not comprehensively evaluated.

pretraining; Sect. 3.3), as well as the robustness of the pretrained controller to out-of-distribution shifts (Sect. 3.4).

## 3.1 PRETRAINING PERFORMANCE

Across three independent trials, each using a distinct dataset of randomly-generated morphologies and environments, pretraining exhibited stable learning trajectories with low variance across trials (Fig. 6A), converging in approximately 1400 learning steps (57 minutes of wall-clock time). Loss was defined as the ratio of final to initial distance from the target light source. At initialization with random controller weights, this ratio was 1.0, indicating robots remained stationary throughout simulation. After pretraining, the loss stabilized at approximately 0.3, representing a 70% improvement. That is, in environments sampled from the training distribution, robots using the pretrained universal controller traversed an average of 70% of their initial distance to the light source. Since each training batch used novel morphologies, we omitted model selection with a validation set.

To visualize the breadth of morphological diversity handled by the pretrained controller, Fig. 11 showcases a representative sample of successful robots. These examples were drawn uniformly from the top-performing 50% of the test morphology set. The sampled bodies exhibit high variation in both body size and shape, demonstrating the non-trivial generalization of the universal controller.

We emphasize that, during pretraining, none of the morphologies used as training samples were ever evaluated. In contrast, zero-shot evolution and few-shot evolution both tested individual morphologies across a set of 10 standardized environments (Appx. A.4). Using the pretrained controller to compare performance of morphologies from the pretraining dataset and morphologies discovered during an application of one of these two algorithms, which employed zero-shot or few-shot design evaluation, would be inherently confounded.

## 3.2 Zero- and few-shot evolution (with pretraining)

A population of morphologies was evolved through random mutation and crossover operations, using the pretrained universal controller. On the same challenging set of tasks used for evaluating pretrained controller generalization, the population converges to near optimal performance in 100 generations of evolution (17 minutes of wall-clock time) without fine-tuning the controller ("zero-shot evolution"; Sect. 2.5). Although zero-shot evolution shows rapid convergence in controlling thousands of distinct bodies, this success masks a key pattern: design population diversity decreases as performance improves. Fig. 6C reveals this pattern—after a brief diversity spike at evolution's onset, the population gradually homogenizes. We term this phenomenon diversity collapse, measuring diversity as the population's mean, pairwise Hamming distance in (and normalized to) the genotype space $\mathcal{G}$ (defined in Sect. 2.1). This metric naturally reflects differences in morphology (body) as well as sensing and actuation masking in the universal controller (brain).

We found that generational fine-tuning of the universal controller for the current population ("few-shot evolution") not only preserves diversity but in fact significantly increases diversity (Fig. 6D). This is a somewhat surprising result as there was no explicit selection pressure to maintain diversity. The process of morphological evolution seems to intrinsically increase population diversity. However, in absence of generational fine-tuning, there is a tipping point at which it is easier to purge diversity, replacing the worst designs with slightly modified clones of the best, than to discover novel morphological innovations with superior performance.

## 3.3 Simultaneous co-design (*without* pretraining)

Ablating pretraining (and fine-tuning), and instead simultaneously optimizing morphology and universal control, together from scratch, results once again in rapid diversity collapse (Fig. 6B). Performance plateaus in well under 180 generations, corresponding to 360 controller learning steps and 109 minutes of wall-clock time. The extent of diversity collapse can be seen in Fig. 12B, where we visualize morphologies from one of the three independent trials, and in Fig. 9 where we plot morphological variance across evolved populations in terms of footprint size and body weight. In both of these analyses, evolved morphologies, and statistics thereof, were extracted from the generation of zero- and few-shot evolution (G31 and G6, respectively) at which average loss matched or beat that achieved by simultaneous co-design (G180).

In all three co-design paradigms (zero-shot, few-shot, simultaneous), universal control enabled successful crossover (Fig. 8). In terms of offspring survival, crossover was initially more successful than mutation. However, in the case of simultaneous co-design, this was not an apples-to-apples comparison because each generation provided the controller with more time to learn how to control the population, and the randomly initialized controller was very bad at the task. Thus, it was not clear if the success of offspring was due to changes in parent morphology or improvements to the universal controller. The performance improvements observed during pretraining, across random morphologies, suggest that the designs produced by crossover during simultaneous co-design may have been no better than random designs. In zero-shot and few-shot evolution, however, the pretrained controller was quite good at the very start, and in zero-shot the controller was not updated during evolution, providing clear evidence of successful crossover prior to diversity collapse (Fig. 8).

## 3.4 Generality of results

While the universal controller was tested out-of-distribution by virtue of morphological evolution, the above simulation-based results ignore many of the practical challenges of physical robots, including the inevitability of component damage and failure. To further assess robustness of the evolved designs and the generality of the pretrained controller to out-of-distribution morphologies, we incre-

mentally disabled their sensors and motors (Fig. 14E). On average, designs retained their evolved functionality with one quarter of their motors failing and more than half of their sensors failing. Next, we tested the ability of zero-shot evolution to generalize to novel environments by replacing the continuously varying terrains seen during pretraining with discrete platforms (Fig. 14B,C). Finally, we swapped the light source and photoreceptors used during pretraining with magnetic fields and magnetoreceptors, changing the task from phototaxis to magnetotaxis (Fig. 14D). In both novel terrain and novel task, zero-shot evolution reshaped the morphologies within the population to meet the new distributions using the pretrained controller, significantly improving upon pretraining performance without fine-tuning.

## 4 DISCUSSION

In this paper, we introduced the large-scale pretraining and fine-tuning of a universal controller using differentiable simulation and demonstrated how this approach accelerates the design of complex robots. The learned controller allows most randomly-generated morphologies (mass-spring networks) to orient along a randomly-generated stimulus (light) vector in three dimensions, and to follow the vector to its source (phototaxis) across challenging, randomly-generated environments (terrains)—more or less: some designs were much better than others, and some outright failed (Fig. 7). Using the pretrained model as a prior, the designer can quickly explore a diversity of changes—from subtle mutations to large recombinations—across arbitrary numbers of distinct designs in parallel without destroying the functionality of working designs, and without constantly readapting the controller to support every morphological innovation.

We intentionally chose a vanilla evolutionary algorithm as "the designer" and a minimal neural architecture for the universal controller to illustrate the power and potential of our approach. We were particularly surprised by the effectiveness of a simple MLP in controlling such large numbers of morphologically complex robots across such challenging terrains. Interestingly, the gaits generated by the universal controller were quite different from those tailored for individual body plans in similar conditions (Strgar et al., 2024); instead of walking or ambling across the rugged terrain, the universal controller discovered patterns of saltation (hopping) not unlike those of kangaroos, in which coordinated actuation of muscles is followed by an aerial phase.

It is important to note, however, that while our controller was universal across the robot's morphology and task environment, we only considered a single material (soft), percept (light), actuator (linear), and task (phototaxis). Extending this approach to multiple tasks that demand more intricate, multi-material body plans with multi-modal sensing (e.g. not just moving toward a single stimulus source, but reacting to various other stimuli, manipulating objects, and working with or against other robots) may require gradually complexifying the neural architecture. This will likely also require replacing the direct genotype-to-phenotype mapping with a more sophisticated (pleiotropic) compression of phenotypes into a latent genome (Li et al., 2025). Instead of presupposing voxel cells with two dozen springs and eight masses, latent genes could control the expression of more atomic building blocks, such as individual masses and springs (or subatomic particles within them), allowing other kinds of non-cubic cells (Hummer & Kriegman, 2024) to emerge. If extended to self-reconfigurable robots, the latent genome or many such genomes may be expressed in myriad ways by a single robot with universal self-control.

We also identified in this paper a previously unknown yet inherent problem of co-designing morphology and universal control—diversity collapse—and showed how to solve this problem through generational fine-tuning. However, this first investigation of diversity collapse only considered a single measure of morphological diversity. Other metrics at both the morphological and behavioral level could be formulated or derived from a latent genotype space. Such metrics could then be incorporated into the design algorithm as a constraint or additional objective.

Another important limitation of this work was that the simulated designs were not transferred to reality. Doing so may require higher resolution simulations (Fig. 13) or improvements to the simulator, e.g. its model of contact, light, and sensing. Adding noise to these models can also ensure that the robot's behavior does not exploit inaccuracies of simulation (Jakobi et al., 1995). The simulator could also be augmented with a neural network that learns the residual physics (Gao et al., 2024). However, the universal controller itself might help reduce the simulation-reality gap since it is, by definition, insensitive to a wide range of variation in the simulated robot's body and world.

Finally, we would like to re-emphasize that results were provided for terrestrial locomotion only, and it remains to determine whether and how the co-design framework introduced here could be extended to other behaviors, such as aquatic, aerial, or arboreal locomotion, as well as manipulation. The latter has been of keen interest in robotics for over a century (Piazza et al., 2019), and, in certain instances, may require more intricate contacts than movement on land. The differentiable contact model used by the simulator in this study could, in principle, be extended to object manipulation, as demonstrated in prior work (Xu et al., 2021; Wang et al., 2023a). Of course, optimizing a single universal controller for an evolving population of distinct effector morphologies may require a more complex neural architecture and, in certain cases, may be intractable. Moreover, the algorithms introduced in this paper require that a differentiable simulator is available or can be created, which may not always be possible.

Despite these limitations, the sheer scale and efficiency achieved by this work opens a new frontier in robot co-design through automatic differentiation, suggesting the breadth of infrastructure and theory developed in fields of deep learning and neural networks may, in some cases, be leveraged by robot co-design in future work.

## ACKNOWLEDGMENTS

This research was supported by NSF awards 2331581 and 2440412, and Schmidt Sciences AI2050 grant G-22-64506.

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

# A APPENDIX

## A.1 ADDITIONAL DISCUSSION

As detailed above in the introduction (Sect. 1), all prior work that trained a universal controller did so in a computationally inefficient manner without recourse to gradient information. And all prior work that leveraged simulation gradients to optimize single-morphology locomotion policies only did so for movement in a perfectly straight line without recourse to external perception. The two algorithmic contributions of this paper are thus: (i) end-to-end differentiable training of a universal controller, which enabled large-scale pretraining and fine-tuning of adaptive sensor-guided behavior; and (ii) new co-design algorithms that utilize this pretrained controller to guide discrete morphological optimization. These novel algorithms established for the first time successful design recombination and revealed the previously unknown phenomenon of diversity collapse, the discovery of which is the primary intellectual contribution of this paper.

Previously, under the computational burden of non-differentiable policy training, others (Huang et al., 2020; Gupta et al., 2022; Wang et al., 2023b; Xiong et al., 2023; 2024; Bohlinger et al., 2024) pretrained a universal locomotion policy but it was unclear whether or how it could be used for morphological evolution. Li et al. (2025) co-designed a population of 64 morphologies and a universal controller but did so simultaneously, which, as we have shown here, can lead to diversity collapse. Wang et al. (2023b) intentionally collapsed diversity to a single morphology after pretraining a universal controller. Schaff & Walter (2022) co-designed a universal controller for two kinds of body plans (quadrupeds and hexapods), optimizing the length and articulation of rigid links within their presupposed spine and leg pairs, which ultimately "converge[d] toward a single design". Morphological diversity matters because some designs generalize better than others to novel circumstances (Fig. 14), the details of which may be impossible to simulate or unknowable a priori.

It may also be helpful to situate our work in relation to that of (Ha, 2019; Schaff et al., 2019; Luck et al., 2020), which used RL to co-optimize the neural control and morphological parameters of a presupposed kinematic tree that could not change during optimization. Although the policy could enlarge or reduce the limbs of "a given morphology", it could not change the overall design: the Ant was always an Ant, with four limbs equally spaced about its torso. Adding and removing body parts was impossible because limb size could not go to zero and new limb buds could not be placed along the Ant. Recombining discrete body parts was impossible because there was not a population of distinct designs to recombine. In short, morphological evolution was impossible. And once again, this prior work treated the simulation as a non-differentiable component of the learning process. Moreover, the resulting controller could only produce blind locomotion in a single, relatively simple body with a small number of degrees of control freedom. Our methods of few-shot and zero-shot evolution, on the other hand, generate a population of thousands of designs with hundreds to thousands of motors that improve performance beyond pretraining in an adaptive, sensor-guided navigation task.

Although discussed above in Sects. 1 and 4, and detailed in Sect. 2.1, it is also worth reiterating that the morphospace used here was limited to actuated springs and sensorized masses on a regular grid, which is consistent with prior work (Cheney et al., 2018; Kriegman et al., 2020a;b; Medvet et al., 2021; Bhatia et al., 2021; Wang et al., 2023b; Li et al., 2024; Strgar et al., 2024; Mertan & Cheney, 2024; 2025). Following Bhatia et al. (2021), the morphospace of many recent co-design studies has been constrained to a small 2D grid of elastic cells in non-differentiable simulation. We hope that the differentiable, 3D simulations released here can provide the basis of a more expressive and scalable benchmark platform for the community.

We stated in Sect. 1 that prior co-design research has mostly focused on simple bodies and behaviors, and one of the goals of our research is to increase complexity in both. The morphologies in this paper have higher motoric complexity (larger numbers of independent motors) than those in prior work, but there are many ways to measure complexity. The endoskeletal robots from Li et al. (2025), for instance, contained fewer motors (and were non-differentiable) but possessed finer grain anatomical complexity (freeform jointed skeletons and soft tissues) than the mass-spring bodies in this paper. We can, to some extent, scale the anatomical complexity of mass-spring robots by simply increasing their number of springs and masses (Fig. 13), but they would still lack the joint constraints and contact models of Li et al. (2025). And while we benchmarked against the simultaneous co-design algorithm from Li et al. (2025), their genetic encoding and evolutionary strategy is almost certainly more scalable than our direct representation and navigation of morphospace. The

algorithms introduced in this paper are inherently synergistic with this and the many other, diverse approaches to robot topology optimization (Wang et al., 2019; Zhao et al., 2020; Gupta et al., 2021; Kriegman et al., 2021a;b; Hu et al., 2022; Matthews et al., 2023; Li et al., 2024; Kobayashi et al., 2024; Yu et al., 2026) including those that also avoid policy retraining for each new morphological variant (Wang et al., 2019; Pathak et al., 2019; Kriegman et al., 2019; Yuan et al., 2022; Schaff & Walter, 2022; Mertan & Cheney, 2024; Lu et al., 2025; Mertan & Cheney, 2025). One of the most promising avenues for future work would be to determine task environments in which it becomes necessary to replace the simple MLP controller used in this paper with one that can better condition behavior on details of the robot's current morphology through masked attention (Kurin et al., 2021; Hong et al., 2022; Gupta et al., 2022; Sferrazza et al., 2024; Li et al., 2025), hypernetworks (Xiong et al., 2024) or contextual modulation (Xiong et al., 2023). Likewise, the minimal evolutionary algorithm used in this paper to optimize the robot's discrete topology may be replaced in future by design algorithms that explicitly promote quality diversity (Lehman & Stanley, 2011; Veenstra & Glette, 2020; Norstein et al., 2023; Mertan & Cheney, 2024; 2025) and thereby strive to realize the creativity and open-endedness of biological evolution in robots.

## A.2 RANDOM ROBOT GENERATION

Random morphologies were generated de novo during pretraining and as the initial seed population of evolution (gen 0). First, we enumerated all possible (length, width, height) tuples with length and width in $[1, 6]$ and height in $[1, 4]$, corresponding to the voxel dimensions of $\mathcal{G}$ (see Sect. 2.1). We then randomly sampled a volume uniformly from the set of possible volumes and subsequently sampled a compatible (length, width, height) tuple to define the bounding box for the genotype. This ensured our dataset contained morphologies of varying volumes and dimensions. Within this bounding box, all voxels were initialized as inactive (zero) and then randomly activated according to probability $p \sim \mathcal{N}(\mu = 0.35, \sigma = 0.125)$, clipped to $[0.1, 0.6]$. If the resulting structure contained no active voxels, sampling was repeated. The largest connected component of active voxels was retained to ensure a valid morphology. If necessary, the bounding box was zero-padded back to $6 \times 6 \times 4$ and the connected component was centered in the horizontal plane and shifted to the bottom of the workspace. The parameters of the sampling distribution were empirically set to produce diverse structures, a sampling of which can be visualized in Figs. 12A and 11.

## A.3 RANDOM ENVIRONMENT GENERATION

Random environments were generated during pretraining (Sect. 2.4), few-shot evolution (Sect. 2.6) and simultaneous co-design (Sect. 2.7). Zero-shot evolution did not require random environment generation since there was no model training involved and thus relied only on evaluation environments (Appx. A.4). An environment consisted of a (terrain, light source position) tuple. A random terrain was generated by sampling a discrete $8 \times 8$ height map of uniformly spaced values. Each value was sampled independently from a Gaussian distribution $\mathcal{N}(\mu = 0, \sigma = \mathcal{U}(0, 0.1))$. A light source position was generated by sampling $(x, y)$ coordinates uniformly inside the circle $(x - r_x)^2 + (y - r_y)^2 = r^2$, where $r \sim \mathcal{U}(0.4, 2.0)$ and $(r_x, r_y)$ was the initial center of mass position of each robot. Prior to the start of simulation, light source positions were placed in 3D by incorporating the terrain height at the sampled (x, y) location.

## A.4 EVALUATION ENVIRONMENTS

During each generation of zero-shot evolution (Sect. 2.5), few-shot evolution (Sect. 2.6) and simultaneous co-design (Sect. 2.7), designs in the population were evaluated on a fixed set of 10 testing environments. This dataset was constructed as follows. Light sources were placed in two rings centered about the robot's starting position: five targets at radius 1.5 and five at radius 2.0, with their angular positions offset to maximize radial coverage. Terrains were sampled at five uniformly spaced difficulty levels, characterized by height map standard deviations in $\{0.02, 0.04, 0.06, 0.08, 0.1\}$. Each ring of light position targets was randomly paired with one terrain from each difficulty level.

### A.5 SENSOR FIELD MODEL

After conducting the experiments in this paper, we identified a discrepancy in our simulation code in which "light" intensity was erroneously modeled as decaying with inverse square root distance, $d^{-1/2}$, rather than the inverse square law, $d^{-2}$. This discrepancy corresponds to a smooth, strictly monotonic, and invertible reparameterization of a physical light source. In particular, a simple transformation of the inverse square law (i.e. raising the signal to the 1/4 power) yields the light model employed in our experiments. The same transformation maps the inverse cube law to our "magnetic field" model (Sect. 3.4; Fig. 14). We verified that, under this transformation, pretrained controller performance remains consistent.

## LLM USAGE

In a few instances, we used LLMs to refine technical language in our Methods section. The sole purpose of this was to improve succinctness and clarity for readers. LLMs only refined original content contributed by the authors and no LLM authored any new content used in this paper. At times, we also used LLM-based coding tools to facilitate rapid prototyping. These tools were particularly useful for generating complex figures. All LLM-generated code was manually verified for correctness.

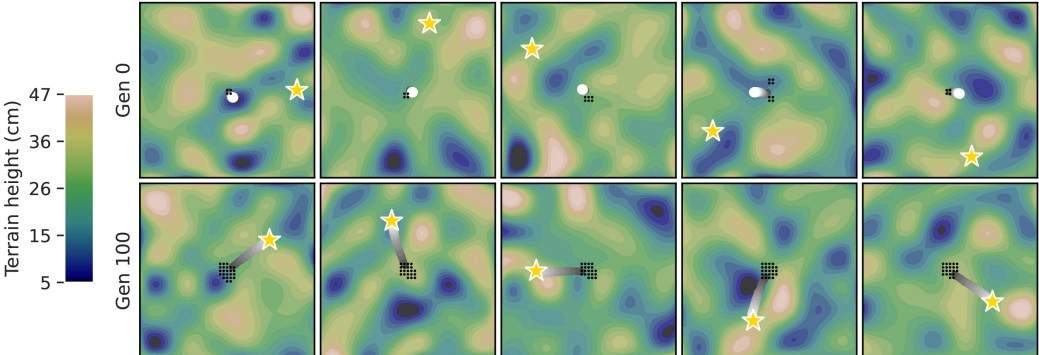

Figure 7: **Evolution of phototaxis.** The five worst designs in the population are depicted before (top row) and after (bottom row) zero-shot evolution. Each design (black dotted footprints) was placed in the center of a randomly generated map. Before evolution, not all of the designs in the population could move (gray to white trajectories) across any terrain toward a light source (gold stars) using the pretrained controller. After evolution, they could. One of the design principles that evolution discovered is that larger footprints increase locomotion stability.

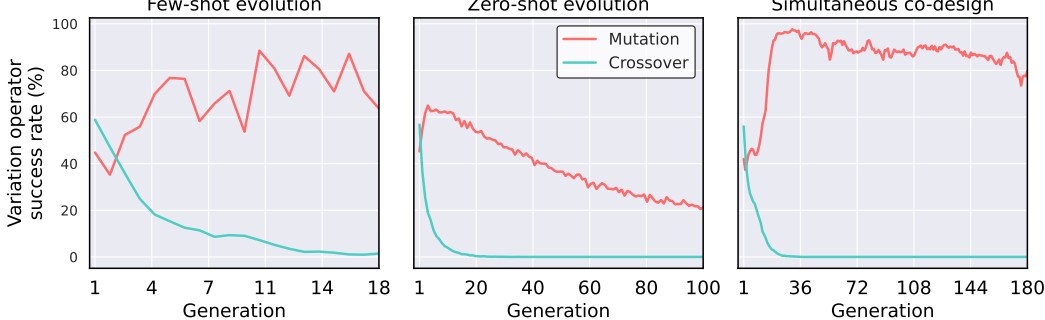

Figure 8: **Success of crossover vs. mutation.** The evolutionary success of mutation and crossover is here defined by the fraction of mutation and crossover events from the previous generation that were absorbed into the current population. Early in evolution, the pretrained controller enables greater than 50% crossover success rate. In the first generation of zero-shot evolution, for instance, 77% of crossover attempts resulted in offspring that were better than at least one of their parents, and more than half of these offspring were better than both of their parents. After a few generations, mutations that finely tune good designs were less likely to be deleterious than exchanging large components between designs.

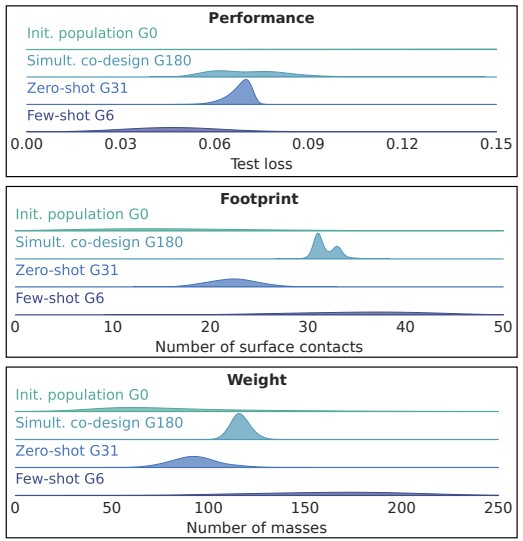

Figure 9: **Evolved populations.** Population performance, phenotype footprint size, and body mass for the initial (randomly generated) and evolved design populations. Whereas zero-shot evolution shifts the population toward smaller designs that are easier to control with the pretrained policy, few-shot evolution maintained a diverse population of overall larger designs with larger footprints which increase locomotion stability.

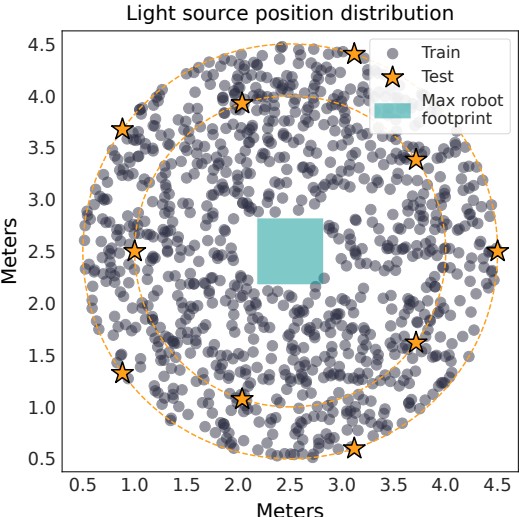

Figure 10: **Phototaxis training and testing.** During pretraining, simultaneous co-design, and few-shot fine-tuning, training light source locations (gray circles) were sampled uniformly within a circle centered on the robot's initial position (blue square). At every learning step, a batch of 8192 randomly positioned lights was sampled, and each was paired with a unique morphology and random terrain. Test light source locations (orange stars) were identical across all methods for fair comparison.

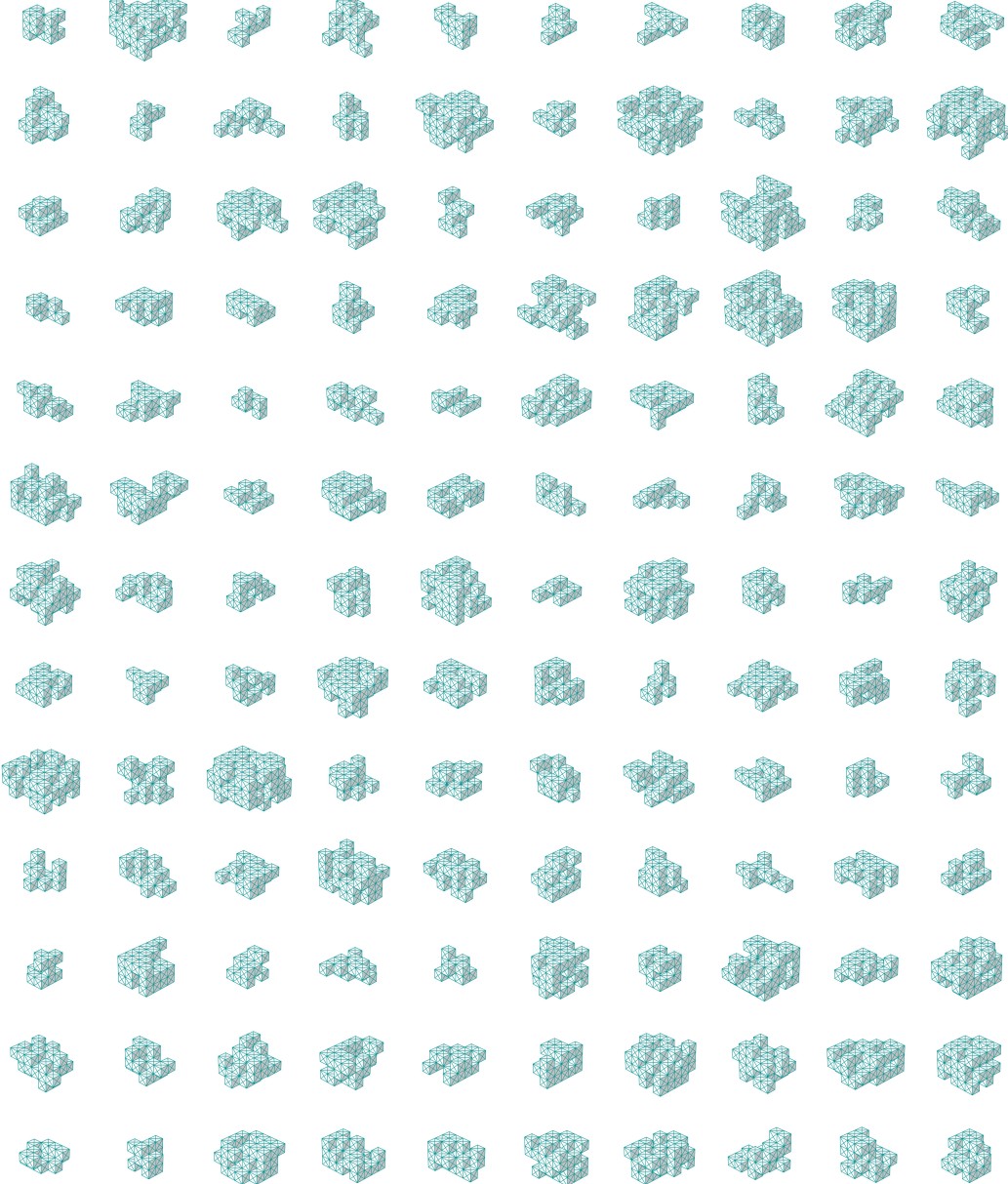

Figure 11: **Generalization of pretrained universal controller.** Randomly sampled morphologies from the top 50% of performers in generation 0 of zero-shot evolution. The universal controller successfully controls these diverse, previously unseen robot designs, demonstrating effective generalization across morphologies.

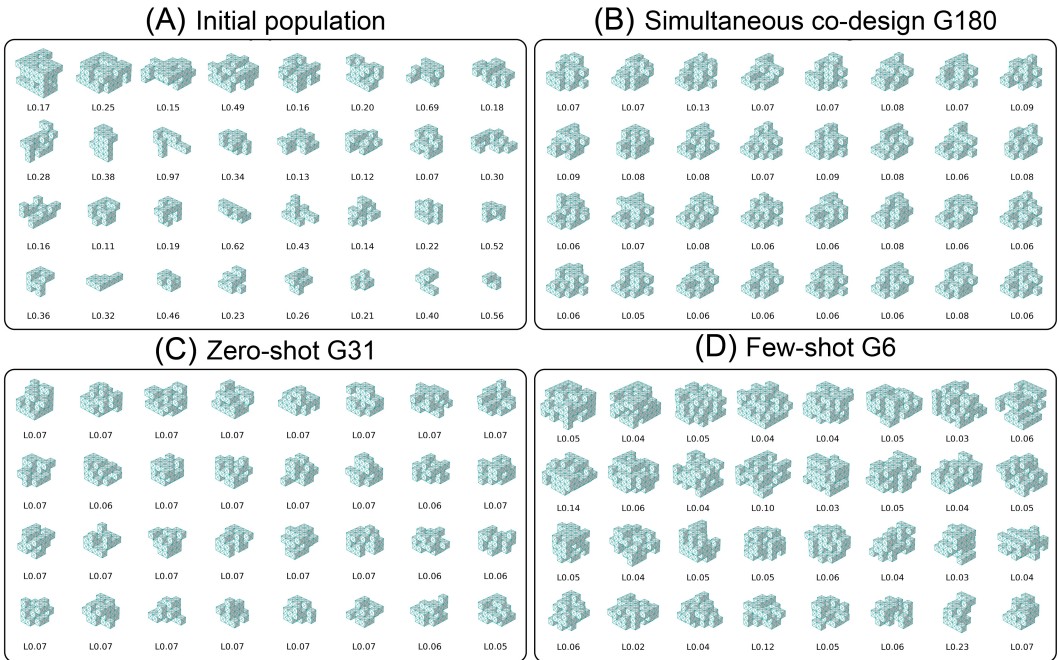

Figure 12: **Morphological distinctiveness.** Robot designs shown are sampled uniformly from each generation's test performance distribution and arranged (left to right, top to bottom) by morphological distinctiveness, defined as the mean pairwise Hamming distance to its peer designs. Performance scores appear below each design. The initial population (**A**) exhibits diverse morphologies with broad performance variation, serving as the starting point for all methods. After 180 generations, simultaneous co-design (**B**) yields high-performing but morphologically homogeneous designs. In contrast, both zero-shot evolution at generation 31 (**C**) and few-shot evolution at generation 6 (**D**) achieve equal or superior performance while maintaining greater morphological diversity and complexity.

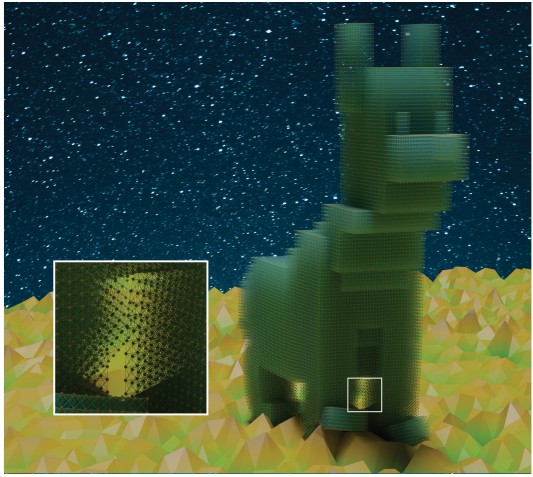

Figure 13: **Scaling morphology.** The embarrassingly parallel nature of the co-design pipeline allows the compute required to simulate 1024 robots with up to 1648 springs (i.e. a single GPU) to be redistributed for a single robot with 1,115,157 springs.

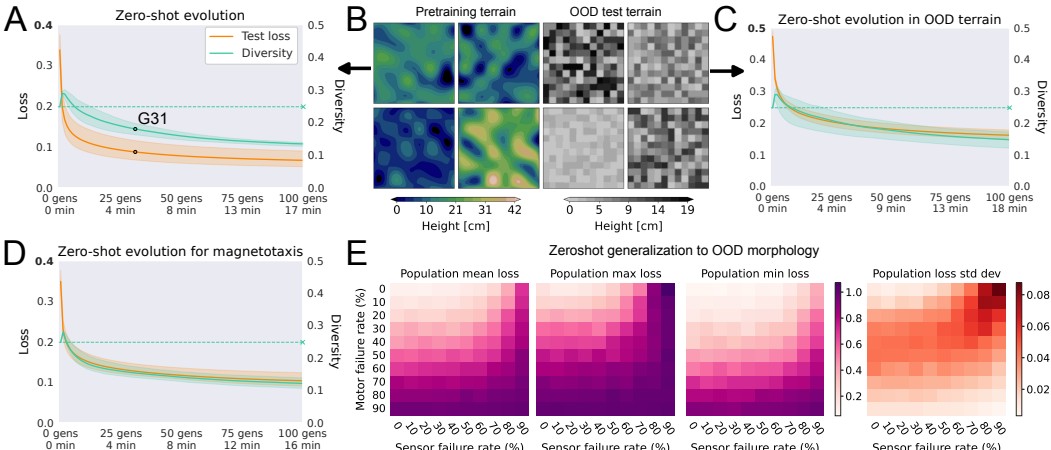

Figure 14: **Zero-shot evolution, out of distribution.** Without any changes to the pretrained controller, zero-shot evolution rapidly improves performance by sculpting the discrete topologies of the robots in the design population through cumulative selection across generations of design mutation and recombination (**A**). This is true whether the distribution of terrains is similar to the continuously varying topographies experienced during pretraining (smooth blue-green maps in **B**) or if terrain distribution is replaced with discrete platforms of randomized height (grayscale checkerboards in B). Using the same pretrained controller, zero-shot evolution generalized to the tested out-of-distribution terrains, significantly improving upon pretraining performance (**C**). Next, we altered the controller's perceptual categories by replacing one kind of sensory input (light) with another (magnetic fields), requiring the policy to generalize from the original behavior optimized during pretraining (photo-taxis) to another (magnetotaxis; **D**). Zero-shot evolution finds designs that "undo" this perceptual shift to once again beat pretraining performance. Finally, we tested the ability of the evolved designs to generalize to out of distribution perceptual and action constraints, without any further evolution or policy fine-tuning (**E**). This last test measures the design's robustness to sensor and motor failure, which is ubiquitous in physical robots. As in Fig. 6, evolved populations were extracted from G31, the generation at which average performance surpassed that of simultaneous co-design, the state-of-the-art benchmark from Li et al. (2025). Morphologies from G31 had their motors and sensors randomly knocked out prior to evaluation. On average, under a simultaneously high rate (60-70%) of sensor failure and a modest rate (20-30%) of motor failure, the population preserves pretraining performance. The performance differential between the best and the worst design in the population highlights the importance of maintaining a population of unique designs. The robustness of the pretrained controller to sensor and motor failure also reflects positively on potential for sim2real transfer where power constraints limit the number of actuating and sensing components and robots must withstand hardware failures.

