# OpenReview forum: "Accelerated co-design of robots through morphological pretraining"
_ICLR.cc/2026/Conference — ICLR 2026 Poster_

### Official Review · Reviewer_MNzL · 2025-10-25

**Soundness:** 2
**Presentation:** 4
**Contribution:** 1
**Rating:** 2
**Confidence:** 5

**Summary:**

This paper presents a framework for accelerating robot morphology–control co-design through morphological pretraining. The authors propose training a universal controller across millions of simulated morphologies using differentiable physics, allowing rapid evaluation of new robot designs without retraining. The pretrained controller is then used in “zero-shot” and “few-shot” evolutionary experiments to explore and refine robot morphologies efficiently. The results show that the pretrained universal controller enables faster convergence and greater morphological diversity than training from scratch.

**Strengths:**

- The paper is clearly written and includes detailed descriptions of both the method and the experimental setup. The authors do a good job explaining their simulation and differentiable modeling pipeline, and the visuals and figures effectively communicate the core ideas and results.
- The motivation for leveraging morphological pretraining is sound, and the idea of a universal differentiable controller that enables rapid design-space exploration is intuitively appealing. The experiments convincingly demonstrate that pretraining can lead to both performance gains and increased design diversity during evolution.

**Weaknesses:**

- Despite the paper’s technical completeness, it reads more as a comprehensive experiment report than as a research contribution with clear conceptual novelty. Most components (differentiable simulation, universal controllers, and evolutionary co-design) are existing ideas in the literature, and it remains unclear what fundamentally new insight or principle this work establishes beyond scaling up prior approaches.
- The lack of comparison against existing co-design or differentiable evolution baselines (e.g., DiffAqua, BodyGen, or Metamorph) makes it difficult to quantify the method’s true advantage. Even a simplified baseline comparison would make the contribution more concrete.
While the concept of “morphological pretraining” is interesting, its novelty is mainly terminological. The paper would benefit from a clearer articulation of how it differs conceptually from pretraining universal controllers in prior work.

**Questions:**

- It would help if the authors clarified what specific insights this framework reveals that earlier differentiable co-design studies missed. For example, does morphological pretraining uncover a new pattern of transfer, scalability, or stability in evolution?
- Additionally, the paper could discuss how this approach might generalize to real hardware or to non-locomotion tasks, given that all experiments are simulation-based.

---

> ### Author Response · Authors · 2025-11-20
>
> We thank the reviewer for their consideration of our work and appreciate that they find strengths in our paper’s presentation and motivation as well as noting its convincing experiments and technical completeness. Below, we respond to the first identified weakness. In subsequent comments we address the other weaknesses and reviewer's questions, and provide references.
>
> ---
>
> **Weakness 1**
>
> "**Despite the paper’s technical completeness, it reads more as a comprehensive experiment report than as a research contribution with clear conceptual novelty. Most components (differentiable simulation, universal controllers, and evolutionary co-design) are existing ideas in the literature, and it remains unclear what fundamentally new insight or principle this work establishes beyond scaling up prior approaches.**"
>
> We appreciate the reviewer’s concern. It is true that the components of our algorithm (differentiable simulation, universal controllers, and evolutionary co-design) are existing ideas in the literature. However, we are the first to combine them successfully, and in doing so we have made what we believe to be an important discovery (diversity collapse) and unlocked an important new capability for evolutionary co-design (recombination). At the end of our paper (Appx. A.1) we restated what we believe to be the two algorithmic contributions of our work:
>
> > “*(i) end-to-end differentiable training of a universal controller, which enabled large scale pretraining and finetuning of adaptive sensor-guided behavior; and (ii) new co-design algorithms that utilize this pretrained controller to guide discrete morphological optimization.*”
>
> There is only one case reported in the literature (Li et al., 2025) in which a universal controller was used to evolve a population of designs; however, they performed these tasks simultaneously, an algorithm we refer to in our paper as simultaneous co-design. In our paper we identify the phenomenon of diversity collapse, which we believe may be inherent to co-designing morphology and universal control. Indeed, in Li et al. (2025), while diversity was not measured and diversity collapse was not shown, it is mentioned that the evolving population often converged to simple two-jointed “snakes”. One of the novel algorithms we presented in our submission, few-shot evolution, solves diversity collapse in a general manner. We thus believe the discovery, articulation, and characterization of diversity collapse to be an important scientific contribution of the paper, in addition to the algorithmic contributions mentioned above. We restate this at the end of the paper:
>
> > “[We] *identified in this paper a previously unknown yet inherent problem of co-designing morphology and universal control—diversity collapse—and showed how to solve this problem through generational finetuning* [few-shot evolution].”
>
> > “[The] *discovery of* [diversity collapse] … *is the primary intellectual contribution of this paper.*”
>
> On the topic of diversity collapse, we would like to point out that the two other anonymous Reviewers, e2bw and 34hu, agreed with our assessment:
>
> > “I think it provides great insight into the use of the generalisation abilities of morphology-agnostic controllers. Especially, shining a light on new open problems, such as the diversity collapse mentioned, is interesting to the community. Overall, I think the paper provides the robot leanring community with further insight into the co-design problem and the use of universal polcies in this setting.”
>
> > “The identification of "diversity collapse" is a key intellectual contribution. It provides a formal explanation for a likely unstated observation in many prior co-design studies.”
>
> Finally, but perhaps most importantly for evolutionary robotics, we here show for the first time successful design recombination (aka crossover) during evolution (Sect. 3.3, Figs. 4, 5, 8). We describe this contribution in the Introduction:
>
> > “[T]*here is a tacit assumption in robotics that crossover—the combining of two parent designs to produce offspring—is so unlikely to produce viable offspring, that it is better to omit crossover altogether … While instances have been reported in which two morphologies were combined using crossover to produce a new morphology (Sims, 1994; Bongard & Pfeifer, 2001; Hiller & Lipson, 2010; Strgar et al., 2024), it was not clear if crossover ever produced offspring with equal or better fitness than either one of their parents*”
>
> And in Figure 8 we report:
>
> > “*Early in evolution, the pretrained controller enables greater than 50% crossover success rate. In the first generation of zero shot evolution, for instance, 77% of crossover attempts resulted in offspring that were better than at least one of their parents, and more than half of these offspring were better than both of their parents.*”
>
> We sincerely hope that this summary has helped clarify for the reviewer the contributions of our work.
>
> ---

---

> ### Author Response · Authors · 2025-11-20
>
> Below we address the second weakness and the first question.
>
> ---
>
> **Weakness 2**
>
> "**The lack of comparison against existing co-design or differentiable evolution baselines (e.g., DiffAqua, BodyGen, or Metamorph) makes it difficult to quantify the method’s true advantage. Even a simplified baseline comparison would make the contribution more concrete. While the concept of “morphological pretraining” is interesting, its novelty is mainly terminological. The paper would benefit from a clearer articulation of how it differs conceptually from pretraining universal controllers in prior work.**"
>
> Respectfully, our paper does compare our approach to a co-design baseline (Li et al., 2025), which we argue is the most appropriate baseline because it is the only prior work to use a universal controller for evolutionary co-design. In Sect. 2.7, we describe the benchmark, and why this baseline does more than simply show empirical performance differences—it also isolates the effect of morphological pretraining. We have copied and pasted the relevant passage below for convenience:
>
> > “*In our third and final experimental group, we remove morphological pretraining and instead simultaneously evolve a population of robots and learn their universal controller, from scratch. This is the algorithm used by Li et al. (2025). Here, it serves as a benchmark of the state of the art (bestowed with simulation gradients for fair comparison) while also isolating the effect of pretraining through ablation. Unlike few-shot evolution, controller parameters and optimizer state are inherited across generations rather than being reset. The genetic algorithm operates as before, but we reduce the per-generation training to just two learning steps (one for parents, one for offspring) to maintain parity with our pretraining experiments, where each training batch was unique.*”
>
> We understand how this important detail can be lost to the methods when viewing Results. So we have edited the caption of Fig. 6 to remind readers that simultaneous co-design is the algorithm from Li et al. (2025).
>
> Our results show that both of the algorithms we propose achieve superior performance compared to the baseline (Fig. 6; Sects. 3.2 and 3.3), and the few-shot evolution algorithm also solves the diversity collapse problem suffered by the baseline (Fig. 6; Sect. 3.2).
>
> The three other methods mentioned by the reviewer as possible baselines are, respectfully, out of scope. It is unclear how the controller architecture employed in either Metamorph (Gupta et al., 2022) or BodyGen (Lu at al., 2025) could be generalized to the mass-spring workspace used in our paper. Moreover, Metamorph is not a co-design algorithm. Finally, Diffaqua (Ma et al., 2021) leverages simulation gradients but is not an evolutionary design algorithm: it requires basis shapes of the same topology to be predefined and then interpolates between them.
>
> As to how our paper differs from pretraining universal controllers in prior work, we are the first to train a universal controller with simulation gradients. Moreover, we are the first to use a pretrained universal controller to accelerate the evolution of robot morphology, and, in doing so, we are the first to identify the inherent pathology of diversity collapse and solve it algorithmically with our method of few-shot evolution. We also kindly refer the reviewer to Appx. A.1 where we provide an in-depth account of related work – and our work’s unique contributions – on topics including universal controller training and morphological topology optimization. Here, we highlight anonymous Reviewer e2bw’s positive comment on our efforts:
>
> > “*The literature review and discussion of related work is comprehensive and extensive, covering all important recent works.*”
>
> We hope this helps clarify our use of an appropriate baseline as well as articulate the novelty of our work compared with prior studies on related topics.
>
> ---
>
> **Question 1**
>
> "**It would help if the authors clarified what specific insights this framework reveals that earlier differentiable co-design studies missed. For example, does morphological pretraining uncover a new pattern of transfer, scalability, or stability in evolution?**"
>
> Here we briefly repeat the contributions and insights described while addressing Weakness 1 and 2. Our framework revealed the previously unknown phenomenon of diversity collapse that is inherent in robot co-design with a universal controller. We also demonstrated how to solve diversity collapse with our algorithm few-shot evolution, which employs generational finetuning of the universal controller. Finally, we show how morphological pretraining enables for the first time in evolutionary robotics successful recombination (crossover) of designs. Please see our responses above for more details.
>
> ---

---

> > ### Author Response · Authors · 2025-11-20
> >
> > Below we respond to the second question and provide references.
> >
> > ---
> >
> > **Question 2**
> >
> > "**Additionally, the paper could discuss how this approach might generalize to real hardware or to non-locomotion tasks, given that all experiments are simulation-based.**"
> >
> > It is true that all our experiments are simulation-based and restricted to locomotion, and we appreciate the reviewer raising the question of the generalizability of our approach. To clarify, we have added the following to our Discussion:
> >
> > > “*Finally, we would like to re-emphasize that results were provided for terrestrial locomotion only, and it remains to determine whether and how the co-design framework introduced here could be extended to other behaviors, such as aquatic, aerial, or arboreal locomotion, as well as manipulation. The latter has been of keen interest in robotics for over a century (Piazza et al., 2019), and in certain instances may require more intricate contacts than movement on land. The differentiable contact model used by the simulator in this study could, in principle, be extended to object manipulation, as demonstrated in prior work (Xu et al., 2021; Wang et al., 2023a). Of course, optimizing a single universal controller for an evolving population of distinct effector morphologies may require a more complex neural architecture and, in certain cases, may be intractable. Moreover, the algorithms introduced in this paper require a differentiable simulator is available or can be created, which may not always be possible.*”
> >
> > In Discussion we also describe how our approach might generalize to real hardware:
> >
> > > “*Another important limitation of this work was that the simulated designs were not transferred to reality. Doing so may require higher resolution simulations (Fig. 13) or improvements to the simulator, e.g. its model of contact, light, and sensing. Adding noise to these models can also ensure that the robot’s behavior does not exploit inaccuracies of simulation (Jakobi et al., 1995). The simulator could also be augmented with a neural network that learns the residual physics (Gao et al., 2024). However, the universal controller itself might help reduce the simulation-reality gap since it is by definition insensitive to a wide range of variation in the simulated robot’s body and world.*”
> >
> > We also extensively tested the universal controller’s generalization to various degrees of sensor and motor failure, known limitations of building and deploying physical machines. In Fig. 14 we describe:
> >
> > > “*On average, under a simultaneously high rate (60-70%) of sensor failure and a modest rate (20-30%) of motor failure, the population preserves pretraining performance. The performance differential between the best and the worst design in the population highlights the importance of maintaining a population of unique designs. The robustness of the pretrained controller to sensor and motor failure also reflects positively on potential for sim2real transfer where power constraints limit the number of actuating and sensing components and robots must withstand hardware failures.*”
> >
> > ---
> >
> > **References**
> >
> > M. Li et al. Generating freeform endoskeletal robots. In *Proceedings of the International Conference on Learning Representations*, 2025.
> >
> > K. Sims. Evolving 3D morphology and behavior by competition. *Artificial Life*, 1994.
> >
> > J. Bongard & R. Pfeifer. Repeated structure and dissociation of genotypic and phenotypic
> > complexity in artificial ontogeny. In *Proceedings of the Genetic and Evolutionary Computation
> > Conference*, 2001.
> >
> > J. Hiller & H. Lipson. Evolving amorphous robots. In *Proceedings of the Conference on
> > Artificial Life*, 2010.
> >
> > L. Strgar et al. Evolution and learning in differentiable robots. In *Proceedings of Robotics: Science and Systems*, 2024.
> >
> > A. Gupta et al. Metamorph: Learning universal controllers with transformers. In *Proceedings of the International Conference on Learning Representations*, 2022.
> >
> > H. Lu et al. BodyGen: Advancing Towards Efficient Embodiment Co-Design. In *Proceedings of the International Conference on Learning Representations*, 2025.
> >
> > P. Ma et al. Diffaqua: A differentiable computational design pipeline for soft underwater swimmers with shape interpolation. *ACM Transactions on Graphics*, 2021.
> >
> > C. Piazza et al. A century of robotic hands. *Annual Review of Control, Robotics, and Autonomous Systems*, 2019.
> >
> > J. Xu et al. An end-to-end differentiable framework for contact-aware robot design. In *Proceedings of Robotics: Science & Systems*, 2021.
> >
> > T.H. Wang et al. Diffusebot: Breeding soft robots with physics-augmented generative diffusion models. In *Advances in Neural Information Processing Systems*, 2023.
> >
> > N. Jakobi et al. Noise and the reality gap: The use of simulation in evolutionary robotics. In *Proceedings of the European Conference on Artificial Life*, 1995.
> >
> > J. Gao et al. Sim-to-real of soft robots with learned residual physics. *Robotics and Automation Letters*, 2024.

---

### Official Review · Reviewer_34hu · 2025-10-27

**Soundness:** 3
**Presentation:** 3
**Contribution:** 3
**Rating:** 6
**Confidence:** 5

**Summary:**

This paper presents a timely contribution to the field of embodied co-design of artificial agents. The core idea is to leverage a process of large-scale, gradient-based "morphological pretraining" in a differentiable simulator to learn a single, universal controller that is agnostic to the robot's morphology. This pretrained controller is then used to accelerate evolutionary design in two novel paradigms: "zero-shot evolution" (using a frozen controller) and "few-shot evolution" (with generational fine-tuning). The authors identify and analyze a critical pathology termed "diversity collapse," where simultaneous co-optimization of morphology and control leads to a homogenization of designs, and demonstrate that their few-shot approach effectively mitigates this issue.

**Strengths:**

1. The identification of "diversity collapse" is a key intellectual contribution. It provides a formal explanation for a likely unstated observation in many prior co-design studies.

2. The scale of the experiments is impressive (e.g., pretraining on 10M+ morphologies, evolving populations of 8192 robots).

**Weaknesses:**

1. The authors proposed the concept of "morphological pretraining". But they actually used a differentiable simulator to pretrain a single controller. This is a little bit confusing. I thought they were going to pre-train a single morphological design like what PreCo (Wang et al. 2023) does.

2. The task (phototaxis/magnetotaxis) is a single, well-defined objective. While complex due to the rugged terrain, it does not require higher-level behaviors like object manipulation, multi-agent interaction, or dealing with multiple, conflicting stimuli.

3. The direct genotype encoding in a grid, while manageable for this study, may not scale efficiently to larger or more complex robots.

**Questions:**

1. Given the critical role of the differentiable simulator, what specific steps or modifications do you believe are most crucial to successfully transfer these co-designed robots to physical hardware?

2. The universal controller excels at a locomotion-based taxis task. Does it still work in manipulation tasks?

3. You show that few-shot evolution preserves diversity without explicit pressure. Could you speculate on the mechanism behind this? Furthermore, for practical design, how would you guide the search towards a specific type of diversity (e.g., prioritizing walking vs. rolling gaits) using your framework?

**Details Of Ethics Concerns:**

None.

---

> ### Author Response · Authors · 2025-11-20
>
> We thank the reviewer for their insightful comments. We also appreciate that they find our work timely and recognize the significance of the diversity collapse phenomena we identify as well as the scale we achieved in our experiments. Below, in this comment, we respond to the first and second identified weaknesses. In subsequent comments we address the third weakness as well as the reviewer’s questions, and also provide references.
>
> ---
>
> **Weakness 1**
>
> "**The authors proposed the concept of "morphological pretraining". But they actually used a differentiable simulator to pretrain a single controller. This is a little bit confusing. I thought they were going to pre-train a single morphological design like what PreCo (Wang et al. 2023) does.**"
>
> We thank the reviewer for highlighting this point of confusion. Our use of the term “morphological pretraining” refers here to pretraining a universal controller across a large dataset of distinct morphologies. This could be used to optimize a single design as in PreCo (Wang et al., 2023) or a population of diverse designs, as we have shown here. To clarify this we have edited the below passage from our Introduction to better summarize our approach at the very start of our manuscript. Note, here we **bold** our edits to show how the passage has changed:
>
> > “*Here we demonstrate a more scalable approach that starts with large-scale morphological pretraining in differentiable simulation and yields a **single** morphology-agnostic controller **capable of** adaptive sensor-guided behavior **across evolving populations of distinct**, complex mass-spring robots with hundreds to thousands of independent motors.*”
>
> We hope that this is helpful. If the reviewer were to share with us any specific language they found confusing we would be glad to amend it.
>
> ---
>
> **Weakness 2**
>
> "**The task (phototaxis/magnetotaxis) is a single, well-defined objective. While complex due to the rugged terrain, it does not require higher-level behaviors like object manipulation, multi-agent interaction, or dealing with multiple, conflicting stimuli.**"
>
> We appreciate the reviewer’s acknowledgement that the task we consider is complex, and we agree that the behavioral objective we optimized in this paper (locomotion toward a stimulus) is well defined. We acknowledged this in Discussion, and we also described steps we believe could be important to extend this work to the three other behaviors the reviewer mentioned (object manipulation, multi-agent interaction, or dealing with multiple, conflicting stimuli). We have copied and pasted the relevant text below for the reviewer’s convenience:
>
> > “... *while this controller was universal across the robot's morphology and task environment, we only considered a single material (soft), percept (light), actuator (linear), and task (phototaxis). Extending* … *to multiple tasks that demand more intricate, multi-material body plans with multi-modal sensing (e.g. not just moving toward a single stimulus source, but reacting to various other stimuli, manipulating objects, and working with or against other robots) may require gradually complexifying the neural architecture* … [and] *compression of phenotypes into a latent genome*”
>
> ---

---

> > ### Author Response · Authors · 2025-11-20
> >
> > Below we address the third weakness and first two questions.
> >
> > ---
> >
> > **Weakness 3**
> >
> > "**The direct genotype encoding in a grid, while manageable for this study, may not scale efficiently to larger or more complex robots.**"
> >
> > We agree that a direct genotype encoding may not scale to larger or more complex robots. We acknowledged this limitation in our Discussion as well as Appx. A.1; however, as we highlight in Discussion, the simplicity of our encoding was a thoughtful choice:
> >
> > > “*We intentionally chose a vanilla evolutionary algorithm as ‘the designer’* … *to illustrate the power and potential of our approach*”
> >
> > It is well known that indirect encodings can aid the evolution of robots (Cheney et al., 2013; Hornby & Pollack, 2001; Bongard & Pfiefer, 2001). In this regard, we find the use of a simple, direct encoding to be a strength of our paper’s experimental design because it better isolates the role of our proposed algorithms, zero-shot and few-shot evolution, in driving effective search. Future work can simply replace our direct representation with an indirect encoding – such as CPPNs (Cheney et al., 2013), grammar-based approaches (Hornby & Pollack, 2001), and gene regulatory networks (Bongard & Pfiefer, 2001) – without making any changes to the learning process, which operates strictly at the level of the phenotype. Since the aforementioned indirect encodings have well defined mutation and crossover operations no further algorithmic innovation would be required to employ them.
> >
> > ---
> >
> > **Question 1**
> >
> > "**Given the critical role of the differentiable simulator, what specific steps or modifications do you believe are most crucial to successfully transfer these co-designed robots to physical hardware?**"
> >
> > We thank the reviewer for this question. It is true that perhaps the most important limitation of the experiments in this paper is that they were conducted entirely in simulation. In our Discussion, we described the steps and modifications that we believe are the most crucial for sim2real transfer. For the reviewer’s convenience we provide the passage below:
> >
> > > “[An] *important limitation of this work was that the simulated designs were not transferred to reality. Doing so may require higher resolution simulations (Fig. 13) or improvements to the simulator, e.g. its model of contact, light, and sensing. Adding noise to these models can also ensure that the robot’s behavior does not exploit inaccuracies of simulation (Jakobi et al., 1995). The simulator could also be augmented with a neural network that learns the residual physics (Gao et al., 2024). However, the universal controller itself might help reduce the simulation-reality gap since it is by definition insensitive to a wide range of variation in the simulated robot’s body and world.*”
> >
> > ---
> >
> > **Question 2**
> >
> > "**The universal controller excels at a locomotion-based taxis task. Does it still work in manipulation tasks?**"
> >
> > This is an important question. To address it specifically we have added to our Discussion. We provide the complete addition below:
> >
> > > “*Finally, we would like to re-emphasize that results were provided for terrestrial locomotion only, and it remains to determine whether and how the co-design framework introduced here could be extended to other behaviors, such as aquatic, aerial, or arboreal locomotion, as well as manipulation. The latter has been of keen interest in robotics for over a century (Piazza et al., 2019), and in certain instances may require more intricate contacts than movement on land. The differentiable contact model used by the simulator in this study could, in principle, be extended to object manipulation, as demonstrated in prior work (Xu et al., 2021; Wang et al., 2023a). Of course, optimizing a single universal controller for an evolving population of distinct effector morphologies may require a more complex neural architecture and, in certain cases, may be intractable. Moreover, the algorithms introduced in this paper require a differentiable simulator is available or can be created, which may not always be possible.*”
> >
> > ---

---

> > > ### Author Response · Authors · 2025-11-20
> > >
> > > Below we respond to the third question and provide references.
> > >
> > > ---
> > >
> > > **Question 3**
> > >
> > > "**You show that few-shot evolution preserves diversity without explicit pressure. Could you speculate on the mechanism behind this? Furthermore, for practical design, how would you guide the search towards a specific type of diversity (e.g., prioritizing walking vs. rolling gaits) using your framework?**"
> > >
> > > We believe that the generational finetuning affords evolution the ability to adapt to out-of-distribution design innovations by honing the controller to make use of previously unseen, but potentially useful, morphological features. This reduces the selection pressure inherent in co-design with a universal controller to collapse population diversity towards the current design(s) best matched to the pretrained model. We also describe this in Sect. 3.2 of the paper, and for the reviewer’s convenience we provide the passage below:
> > >
> > > > “*The process of morphological evolution seems to intrinsically increase population diversity. However, in absence of generational finetuning, there is a tipping point at which it is easier to purge diversity, replacing the worst designs with slightly modified clones of the best, than to discover novel morphological innovations with superior performance.*”
> > >
> > > For practical design, quality-diversity algorithms could be used to guide search toward specific types of diversity. In Appx. A.1 we provide references to relevant methods:
> > >
> > > > “... *the minimal evolutionary algorithm used in this paper to optimize the robot’s discrete topology may be replaced in future by design algorithms that explicitly promote quality diversity (Lehman & Stanley, 2011; Veenstra & Glette, 2020; Norstein et al., 2023; Mertan & Cheney, 2024; 2025)*”
> > >
> > > Finally, in regards to how one could adapt our approach to favor a specific gait, the simplest adjustment would be to modify the learning objective so that, instead of optimizing only for phototaxis, certain behavioral signatures are also encouraged or penalized. Whereas the phototaxis objective considers only the change in distance between the robot’s center of mass and the target from the first to the last timestep of the episode, one could incorporate other parts of the full trajectory (e.g., positions, velocities, and terrain contacts of each mass point at each timestep) to encourage imitation of a desired gait.
> > >
> > > ---
> > >
> > > **References**
> > >
> > > Y. Wang et al. Preco: Enhancing generalization in co-design of modular soft robots via brain-body pretraining. In *Proceedings of the Conference on Robot Learning*, 2023.
> > >
> > > N. Cheney et al. Unshackling Evolution: Evolving Soft Robots with Multiple Materials and a Powerful Generative Encoding. In *Proceedings of the Genetic and Evolutionary Computation Conference*, 2013.
> > >
> > > G. Hornby & J. Pollack. The Advantages of Generative Grammatical Encodings for Physical Design. In *Proceedings of the Congress on Evolutionary Computation*, 2001.
> > >
> > > J. Bongard & R. Pfeifer. Repeated structure and dissociation of genotypic and phenotypic
> > > complexity in artificial ontogeny. In *Proceedings of the Genetic and Evolutionary Computation Conference*, 2001.
> > >
> > > N. Jakobi et al. Noise and the reality gap: The use of simulation in evolutionary robotics. In *Proceedings of the European Conference on Artificial Life*, 1995.
> > >
> > > J. Gao et al. Sim-to-real of soft robots with learned residual physics. *Robotics and Automation Letters*, 2024.
> > >
> > > C. Piazza et al. A century of robotic hands. *Annual Review of Control, Robotics, and Autonomous Systems*, 2019.
> > >
> > > J. Xu et al. An end-to-end differentiable framework for contact-aware robot design. In *Proceedings of Robotics: Science & Systems*, 2021.
> > >
> > > T.H. Wang et al. Diffusebot: Breeding soft robots with physics-augmented generative diffusion models. In *Advances in Neural Information Processing Systems*, 2023.
> > >
> > > J. Lehman & K. Stanley. Evolving a diversity of virtual creatures through novelty search and local competition.  In *Proceedings of the Genetic and Evolutionary Computation Conference*, 2011.
> > >
> > > F. Veenstra & K. Glette. How different encodings affect performance and diversification when evolving the morphology and control of 2D virtual creatures. In *Proceedings of the Conference on Artificial Life*, 2020.
> > >
> > > E. Norstein et al. Effects of compliant and structural parts in evolved modular robots. In *Proceedings of the Conference on Artificial Life*, 2023.
> > >
> > > A. Mertan & N. Cheney. Towards multi-morphology controllers with diversity and knowledge distillation. In *Proceedings of the Genetic and Evolutionary Computation Conference*, 2024.
> > >
> > > A. Mertan & N. Cheney. Controller distillation reduces fragile brain-body co-adaptation and enables migrations in map-elites. *arXiv preprint arXiv:2504.06523*, 2025.
> > >
> > > ---

---

### Official Review · Reviewer_e2bw · 2025-11-01

**Soundness:** 4
**Presentation:** 3
**Contribution:** 3
**Rating:** 6
**Confidence:** 5

**Summary:**

The paper investigates the co-design of embodiment and control using a pretrained controller architecture. To this end, a policy was trained on 10 million morphologies - the resulting policy was then used to optimise the embodiment of the agent.
The paper investigates the use of such morphology-agnostic control policies when optimising the embodiment of agents in a zero-shot or few-shot manner.
While the technical novelty of the paper is reasonable, I think it provides great insight into the use of the generalisation abilities of morphology-agnostic controllers. Especially, shining a light on new open problems, such as the diversity collapse mentioned, is interesting to the community. Overall, I think the paper provides the robot leanring community with further insight into the co-design problem and the use of universal polcies in this setting.

**Strengths:**

- Interesting control problem, here a population of robots with different morphologies and configurations. Use of new technologies, such as differentiable simulators for fast design optimisation.
- The investigated morphology space is very large and complex - this is a clear strenght of this paper!
- Another particular strength of the paper is providing more insights into the general co-design problem with universal policies by providing deeper analysis.
- The literature review and discussion of related work is comprehensive and extensive, covering all important recent works.

**Weaknesses:**

- The method only considers one simulation environment.
- In Figure 6, it would have been interesting to see the best performance from all morphologies used in the pretraining stage (ie the best out of 10 million). It's not clear to me whether a better design can be found than through this massive grid search.
- An open question is also if the results can be translated to other robotics domains, or if no differentiable simulator is available. This could restrict the applicability of the methods and/or the generalizibility of the discussion points.

**Questions:**

See my question above about the best morphology out of the pretraining dataset.

---

> ### Author Response · Authors · 2025-11-20
>
> We thank the reviewer for their thoughtful comments. We also appreciate that they recognize several strengths of our work including the use of differentiable simulation and large, complex design spaces as well as insights into the previously overlooked challenge of robot co-design with universal controllers. Below, in this comment, we address the first and third identified weaknesses. In a second comment we address the second weakness and the reviewer's only question, which are related, and provide references.
>
> ---
>
> **Weakness 1**
>
> "**The method only considers one simulation environment.**"
>
> It is true that our experiments focus on a single overall task (terrestrial phototaxis). We acknowledged this limitation in the Discussion, and we also outlined what we believe would be required to extend the framework to more (and more complex) tasks:
>
> > “... *while the controller was universal across the robot's morphology and task environment, we only considered a single … task (phototaxis). Extending this approach to* … *multi-modal sensing (e.g. not just moving toward a single stimulus source, but reacting to various other stimuli, manipulating objects, and working with or against other robots) may require gradually complexifying the neural architecture* … [and] *compression of phenotypes into a latent genome*”
>
> However, we would like to emphasize that although the task is fixed, the simulation environment itself is dynamic. Specifically, the environment involves significant terrain variability, and each of the distinct morphologies could, in some sense, be considered a distinct task “environment” from the perspective of the controller. We also thoroughly tested the generality of our controller to various out-of-distribution environmental shifts (Sect. 4.3; Fig. 14). These include perceptual category shift (from phototaxis to magnetotaxis), high rates of random sensor and motor failure, as well as categorically out of distribution terrains. In all three cases, the controller maintained strong performance (Sect. 4.3; Fig. 14), indicating that—even within a single nominal task—the learned policy transfers robustly across substantial environmental and morphological variation. Thus, while broader multi-task evaluation is an important direction for future work, the current study does in fact demonstrate meaningful generalization across stimulus, morphology, and terrain conditions.
>
> ---
>
> **Weakness 3**
>
> "**An open question is also if the results can be translated to other robotics domains, or if no differentiable simulator is available. This could restrict the applicability of the methods and/or the generalizibility of the discussion points.**"
>
> We agree that the applicability of our methods to other robotics domains is an open question, and that a differentiable simulator is necessary. To clarify this we have added a new paragraph to our Discussion, which we provide below for the reviewer’s convenience:
>
> > “*Finally, we would like to re-emphasize that results were provided for terrestrial locomotion only, and it remains to determine whether and how the co-design framework introduced here could be extended to other behaviors, such as aquatic, aerial, or arboreal locomotion, as well as manipulation. The latter has been of keen interest in robotics for over a century (Piazza et al., 2019), and in certain instances may require more intricate contacts than movement on land. The differentiable contact model used by the simulator in this study could, in principle, be extended to object manipulation, as demonstrated in prior work (Xu et al., 2021; Wang et al., 2023a). Of course, optimizing a single universal controller for an evolving population of distinct effector morphologies may require a more complex neural architecture and, in certain cases, may be intractable. Moreover, the algorithms introduced in this paper require a differentiable simulator is available or can be created, which may not always be possible.*”
>
> Reflecting on the reviewer’s question we have also softened our closing remarks in the Discussion by noting that the benefits of our approach may only apply  “*in some cases*”.
>
> ---

---

> ### Author Response · Authors · 2025-11-20
>
> Below we address the second weakness and the reviewer's only question, and provide references.
>
> ---
>
> **Weakness 2**
>
> "**In Figure 6, it would have been interesting to see the best performance from all morphologies used in the pretraining stage (ie the best out of 10 million). It's not clear to me whether a better design can be found than through this massive grid search.**"
>
> We apologize if this was unclear, but during pretraining morphologies were not evaluated. Each design in the pretraining dataset, paired with a random terrain and target light location, was used exactly once as a single training sample. Moreover, the controller at this stage is untrained – therefore the model’s performance on any given morphology was not yet defined.
>
> By contrast, zero-shot evolution and few-shot evolution both rely on explicit, comprehensive task evaluation, where each morphology is tested across a set of 10 standardized terrain-target pairs. To evaluate the pretraining dataset of designs in a comparable way would require running the trained controller on every pretraining morphology across all 10 test environments. Over 30M+ morphologies (10M+ per trial for three trials), this evaluation would require an order of magnitude more compute time than zero-shot evolution, which we feel renders the comparison somewhat uninformative. Most importantly, such a comparison would not be “apples to apples”: pretraining morphologies would be evaluated in-distribution (they were seen during pretraining), whereas morphologies discovered during evolution are evaluated strictly zero-shot or few-shot, depending on the algorithm. Thus, the two sets of scores would not be directly comparable.
>
> We want to thank the reviewer for highlighting this ambiguity as it affects both the clarity of our methods and interpretation of our results. We believe we have addressed the issue through new explanations added to Results, which we feel have substantially improved the paper. For the reviewer’s convenience, we provide the additions below. In Sect. 3.1 we added the following new text:
>
> > “*We emphasize that, during pretraining, none of the morphologies used as training samples were ever evaluated. In contrast, zero-shot evolution and few-shot evolution both tested individual morphologies across a set of 10 standardized environments (Appx. A.4). Using the pretrained controller to compare performance of morphologies from the pretraining dataset and morphologies discovered during an application of one of these two algorithms, which employed zero-shot or few-shot design evaluation, would be inherently confounded.*”
>
> And in the caption for Fig. 6 we added the following:
>
> > “*Note that pretraining loss (blue curve in A) is not directly comparable to co-design performance (B-D) because designs in the pretraining dataset only served as single-use training samples and were not comprehensively evaluated.*”
>
> ---
> **Question 1**
>
> "**See my question above about the best morphology out of the pretraining dataset.**"
>
> We believe we have addressed this question above, and we are glad to have been able to clarify how and when morphologies undergo performance evaluation. We thank the reviewer again for highlighting this question as the resulting additions to the paper have improved our manuscript significantly.
>
> ---
>
> **References**
>
> C. Piazza et al. A century of robotic hands. *Annual Review of Control, Robotics, and Autonomous Systems*, 2019.
>
> J. Xu et al. An end-to-end differentiable framework for contact-aware robot design. In *Proceedings of Robotics: Science & Systems*, 2021.
>
> T.H. Wang et al. Diffusebot: Breeding soft robots with physics-augmented generative diffusion models. In *Advances in Neural Information Processing Systems*, 2023.
>
> ---

---

### Author Response · Authors · 2025-12-02
**Author's summary for the AC**

We thank the AC for their time reviewing our submission. For the AC’s convenience, we provide a succinct summary of each reviewer’s concerns and well as our corresponding clarifications and revisions.

---

**Reviewer e2bw**

This reviewer was broadly positive about our contribution, noting the paper “provides great insight”,  “[shines] a light on new open problems”, and “is interesting to the community”. They commended the scale of our design space and the thoroughness of our literature review.

We think this reviewer’s primary concern was that we omitted reporting the performance of the best morphologies used during pretraining, a potentially very strong baseline. It appears this concern stemmed from a misunderstanding of our methods: the morphologies in the pretraining dataset were never evaluated. As we explained in our rebuttal, doing so would require ~10x the compute time of our algorithms, and comparing in-distribution pretraining morphologies to zero- or few-shot evolved ones would be uninformative. We clarified this with additions to our manuscript (Sect. 3.1; Fig. 6), which we believe would have led the reviewer to raise their score.

The reviewer pointed out that our work considered only a single simulation environment. We clarified that, while we only considered a single overall task, our simulation environments vary extensively in terrain, robot morphology, and perceptual category (Sect. 3.4; Fig. 14). We also augmented our manuscript (Sect. 4) to address questions regarding the generalizability of our method to other robotics domains. For details on both these topics, please see our original rebuttal.

---

**Reviewer 34hu**

This reviewer considered our work a strong contribution, highlighting the paper’s “identification of diversity collapse” and “explanation for a likely unstated observation in many prior co-design studies”. The reviewer appreciated the “impressive” scale we achieved in our experiments and noted that our chosen task was “complex”.

We believe this reviewer was primarily concerned with the generalizability of our method to real hardware and the manipulation task domain. We clarified that our Discussion (Sect. 4) details steps that may aid sim2real transfer. We also highlighted that our universal controller generalized to high rates of sensor failure and modest rates of motor failure, a property directly relevant to physical robustness (Sect. 3.4; Fig. 14). Finally, we added a paragraph to Discussion (Sect. 4) describing how the framework may be extended to manipulation.

The reviewer also stated initial confusion interpreting our method in relation to PreCo (Wang et al. 2023), pointed out that the direct encoding may not scale well, and inquired into the mechanisms underlying diversity preservation as well as extensions thereof. We addressed these concerns and questions, referring to and amending our manuscript as necessary. Please see our rebuttal for details.

---

**Reviewer MNzL**

Despite providing a low overall rating, this reviewer praised the paper’s “technical completeness” and its presentation, rating it as “excellent”. The reviewer noted the intuitive rationale for our methods as well as the persuasiveness of our experiments.

We believe this reviewer was primarily concerned with (1) identifying the conceptual novelty of our work and (2) the inclusion of an appropriate baseline. To address the former, we clarified that our work was the first to train a universal controller using simulation gradients and employ this controller in novel algorithms for accelerated robot co-design. In doing so, we identified (and later solved) the previously unknown phenomenon of diversity collapse. Both other anonymous reviewers confirmed this central and valuable contribution of our paper. We also clarified that our paper demonstrates successful design recombination (i.e., crossover), a first in evolutionary robotics.

Regarding the apparent missing baseline, it seems the reviewer’s concern stemmed from a misunderstanding: our baseline is the simultaneous co-design method of Li et al. (2025) (Sect. 2.7). We explained this in our response and updated the caption of Fig. 6 to make the comparison more explicit. Incidentally, to clarify our contributions over prior work pretraining universal controllers, we referred to our additional discussion of related work (Appx. A.1). Here, we emphasize that Reviewer e2bw found our literature review to be “comprehensive and extensive”.

With the explanations of our paper’s conceptual and algorithmic novelty as well as the clarification of our chosen baseline, we believe that this reviewer would have raised their score. For additional details, we kindly refer the AC back to our original rebuttal, where we also addressed the reviewer's other questions regarding generalizing our method to real hardware and the manipulation task domain.

---

---

### Meta-Review · Area_Chair_uxf1 · 2026-01-08

**Summary:**

The paper proposes a robot co-design framework that utilizes large-scale morphological pretraining to learn a single, morphology-agnostic controller using gradient-based optimization with a differentiable simulator. Rather than learning a bespoke controller for each candidate design, the framework pretrains a universal controller across a large number of morphologies and environments. The framework then uses this pretrained controller to enable users to immediately evaluate the effects of making non-differentiable changes to the morphology, which the paper terms as "zero-shot evolution". Compared to the alternative of simultaneously optimizing the controller while evolving the design, which the paper shows to be prone to "diversity collapse", zero-shot evolution results in more diverse, higher-performing designs.

The paper was evaluated by three reviewers who agree on several of the paper's key strengths and weaknesses. Among the strengths, two of the reviewers emphasize the paper's contributions with regard to providing further insights into the challenges of jointly optimizing robot designs together with an evolving policy. As Reviewer 34hu points out, the identification of diversity collapse as a failure mode of traditional co-optimization problems helps to provide a formal explanation for a phenomenon that has not been well understood. Related, Reviewer MNzL finds the motivation for utilizing morphological pretraining to be sound and the use of a differentiable controller to be appealing. Additionally, at least two reviewers appreciate the size and complexity of the morphology space that the paper investigates. At the same time, reviewers raised several concerns with the paper as originally submitted. Chief among them are questions about the generality of the results, considering that the experiments are limited to a single simulation environment, that the framework relies upon access to a differentiable simulator, and questions regarding the scalability of the genotype encoding. Additionally, Reviewer MNzL feels that the the components that make up the proposed framework already exist and that it is not clear what new insight or principle the paper provides besides scaling up these existing approaches. Related, the reviewer notes that the paper would benefit from a clearer articulation of the framework differs conceptually from earlier work that pretrained universal controllers. Similarly, Reviewer MNzL comments that the paper lacks comparisons to relevant co-design or differentiable baselines, which makes it difficult to evaluate the benefits of the proposed framework.

**Reviewer Concerns:**

The authors made an honest effort to address these concerns in their rebuttal and through updates to the paper. In response to Reviewer MNzL's concerns about the significance of the contributions, the authors acknowledge that the individual components are not new, but emphasize that this is the first work to effectively combine them. Additionally, the authors stress the contributions of the demonstrated diversity collapse phenomenon, something that at least one other reviewer points to as one of the paper's key strengths. Considering the concern about baseline comparisons, the authors clarify that they compare to recent work that utilizes a universal controller to evolve a population of designs. Meanwhile, the authors acknowledge that the evaluation is limited to a single domain, but stress the variability in the terrain and the large and complex space of morphologies.

**Reviewer Scores:**

The AC agrees with the concern raised by Reviewers e2bw and 34hu that the significance of the work is limited by the paper's consideration of a single simulated task, and is not entirely convinced by the rebuttal in this regard. However, the AC feels that the rebuttal sufficiently addressed most of the other questions/concerns that Reviewers e2bw and 34hu bring up. Meanwhile, the AC finds the response to Reviewer MNzL to be convincing, and believes that it warrants an increase in their overall score.

---

### Decision · Program_Chairs · 2026-01-26

Accept (Poster)